# Stanford-ORB: A Real-World 3D Object Inverse Rendering Benchmark

**Zhengfei Kuang**\* **Yunzhi Zhang**\* **Hong-Xing Yu**

**Samir Agarwala** **Shangzhe Wu** **Jiajun Wu**

Stanford University

## Abstract

We introduce *Stanford-ORB*, a new real-world 3D Object inverse Rendering Benchmark. Recent advances in inverse rendering have enabled a wide range of real-world applications in 3D content generation, moving rapidly from research and commercial use cases to consumer devices. While the results continue to improve, there is no real-world benchmark that can quantitatively assess and compare the performance of various inverse rendering methods. Existing real-world datasets typically consist only of the shape and multi-view images of objects, which are not sufficient for evaluating the quality of material recovery and object relighting. Methods capable of recovering material and lighting often resort to synthetic data for quantitative evaluation, which on the other hand does not guarantee generalization to complex real-world environments. We introduce a new dataset of real-world objects captured under a variety of natural scenes with ground-truth 3D scans, multi-view images, and environment lighting. Using this dataset, we establish the first comprehensive real-world evaluation benchmark for object inverse rendering tasks from in-the-wild scenes and compare the performance of various existing methods. All data, code, and models can be accessed at `https://stanfordorb.github.io/`.

## 1 Introduction

When we take a photograph of an object, light travels through space, bounces off surfaces, and eventually reaches the camera sensors to create an image. Inverting this image formation process is a fundamental problem in computer vision—from just the resulting single 2D image, we aim to reconstruct the physical scene in 3D, inferring 3D object shape, reasoning about its reflectance, and disentangling its intrinsic appearance from illumination effects. This process is often referred to as *inverse rendering*. Enabling machines to do so is not only crucial for general image understanding and object analysis, but also facilitates numerous downstream applications, such as controllable 3D content generation [65, 50, 12, 91, 52] and robotic manipulation [77, 51, 20, 14, 66, 61].

Recent years have witnessed significant progress in inverse rendering, with the emergence of various differentiable and neural rendering techniques. Neural Radiance Fields (NeRFs) [43], for instance, have emerged as a powerful volumetric representation that effectively encodes the geometry and view-dependent appearance of a scene into a neural network, allowing for high-fidelity novel view synthesis of complex real-world scenes given raw multi-view images as input. This framework has been successfully extended to further decompose the appearance into explicit material representations (*e.g.*, BRDFs) and environment lighting [63, 88, 9, 7]. As the visual granularity of these inverse rendering results continues to improve, it becomes an increasingly pressing challenge to quantify and compare the performance of different methods.

---

\*Equal contribution.

37th Conference on Neural Information Processing Systems (NeurIPS 2023) Track on Datasets and Benchmarks.

Quantitative evaluation is particularly challenging for inverse rendering tasks because obtaining ground-truth data for material and lighting is prohibitively difficult in practice. Many existing works, therefore, resort to synthetic data for quantitative evaluation [30, 63, 7, 85, 88, 44, 74]. However, evaluation using synthetic data has several critical drawbacks. First, despite the constant evolution of 3D modeling and graphics rendering engines, it remains challenging to date to simulate intricate lighting effects that simply appear in the natural world, *e.g.*, light bouncing inside a pitcher made of shiny metal. Second, synthetic rendering engines typically assume a perfect image formation model, omitting all the noise in a real-world imaging process, which abolishes the generalization guarantee to real-world scenarios. More fundamentally, such synthetic evaluation protocols inevitably give an advantage to methods that exploit similar assumptions that were used to generate the evaluation data.

Existing real-world evaluation benchmarks, on the other hand, are limited on two fronts. Many of them are designed primarily to evaluate geometry and novel view synthesis and only include multi-view images and 3D captures of the objects, without access to the underlying material or lighting properties [26, 28, 7]. Datasets that provide ground-truth lighting and relit images for material and relighting evaluation [29, 67], however, are typically captured in constrained environments and do not represent the rich lighting variations in the natural world.

In this work, we aim to bridge this gap in real-world inverse rendering evaluation by introducing **Stanford-ORB**, a new Real-World 3D **O**bject Inverse **R**endering **B**enchmark. To do this, we collect a new real-world dataset of 14 objects captured under 7 different in-the-wild scenes, with ground-truth 3D scans, dense multi-view images, and environment lighting. This allows us to evaluate different aspects of inverse rendering methods through a number of metrics, including shape estimation, in-the-wild relighting with ground-truth lighting, and novel view synthesis. We compare a wide range of inverse rendering methods using this benchmark, including classical optimization-based methods and learning-based methods with neural representations. All data, trained models, and the evaluation protocol will be released to facilitate future work on this topic.

## 2 Related Work

**State of the Art on Inverse Rendering.** Early works study the interaction of object shape, material, and lighting to estimate individual components, such as shape from shading [6, 87, 2], material acquisition [31, 32, 48, 47, 78], and lighting estimation [73, 80], or to recover reflectance and illumination assuming known shapes [36, 37]. Full-fledged inverse rendering methods aim at estimating all components simultaneously [40], which can be achieved with differentiable renderers [39, 13, 11]. Neural representations have recently emerged as a powerful approach in many state-of-the-art inverse rendering methods, due to their continuous nature and capacity to model complex geometry and appearance. Volumetric representations such as Neural Radiance Fields (NeRFs) [43] and the like [4, 7, 9, 8, 88, 54, 81] encode geometry and appearance as volumetric densities and radiance with a Multi-Layer Perceptron (MLP) network, and render images using the volume rendering equation [42]. While permitting flexible geometry, volumetric representations often suffer from over-parameterization, resulting in artifacts like floaters.

Other surface-based representations [69, 85, 84, 44, 89, 24, 75, 64] extract surfaces as the zero level set, for instance, of a signed distance function (SDF) or an occupancy field [46], allowing them to efficiently model the appearance on the surface with an explicit material model, such as using bidirectional reflectance distribution functions (BRDFs). This also enables modeling more complex global illumination effects, such as self-shadows. Most of these methods focus on per-scene optimization and require dense multiple views as input. Recently, researchers have incorporated learning-based models, distilling priors from large training datasets for fast inference on limited test views [59, 25, 32, 5, 10, 76, 74, 90].

Inverse rendering tasks have also been explored at a scene level beyond single objects. For indoor scenes, existing work has created large synthetic datasets, such as OpenRooms [33], and trained supervised models to predict geometry, material, and spatially-varying lighting with ground-truth data [30, 71, 92]. For outdoor scenes, others have used time-lapse images or Internet photos and trained models with weak supervision from optimization [83, 82, 34]. In this work, we focus on evaluating object-centric inverse rendering tasks.

**Existing Inverse Rendering Benchmarks.** Evaluating inverse rendering results is challenging because collecting ground-truth data is difficult. Table 1 summarizes existing datasets for object

Table 1: Comparison with Existing Object-centric Inverse Rendering Datasets. *Objaverse [19] consists of both synthetic objects and real scans.

| Dataset | # Scenes | # Objects | Real | Scene Type | Multi-view | Shape | Relit Image | Lighting |
|---|---|---|---|---|---|---|---|---|
| ShapeNet-Intrinsics [59] | 98 | 31K | ✗ | synthetic | ✓ | ✓ | ✓ | ✓ |
| NeRD Synthetic [7] | 30 | 3 | ✗ | synthetic | ✓ | ✓ | ✓ | ✓ |
| ABO [16] | 40K | 8K | ✗ | synthetic | ✓ | ✓ | ✓ | ✓ |
| MIT Intrinsics [23] | 1 | 20 | ✓ | studio | ✓ | ✗ | ✗ | ✗ |
| DTU-MVS [26] | 1 | 80 | ✓ | studio | ✓ | ✓ | ✗ | ✗ |
| Objaverse [19] | 818K | 818K | (✓)* | studio | ✓ | ✓ | ✗ | ✗ |
| DiLiGenT-MV [29] | 1 | 5 | ✓ | studio | ✓ | ✓ | ✓ | ✓ |
| ReNe [67] | 1 | 20 | ✓ | studio | ✓ | ✗ | ✓ | ✓ |
| OpenIllumination [35] | 155 | 64 | ✓ | studio | ✓ | ✗ | ✓ | ✓ |
| Lombardi et al. [36] | 5 | 6 | ✓ | in-the-wild | ✗ | ✓ | ✓ | ✓ |
| NeRD Real [7] | 4 | 4 | ✓ | in-the-wild | ✓ | ✗ | ✓ | ✗ |
| NeROIC [28] | 10 | 3 | ✓ | in-the-wild | ✓ | ✗ | ✓ | ✗ |
| Oxholm et al. [47] | 3 | 4 | ✓ | in-the-wild | ✓ | ✓ | ✓ | ✓ |
| Stanford-ORB (ours) | 7 | 14 | ✓ | in-the-wild | ✓ | ✓ | ✓ | ✓ |

inverse rendering evaluation. The early MIT Intrinsics dataset [23] provides a small dataset of intrinsic images of real objects, including albedo and shading maps obtained through polarization techniques [45]. Bell et al. [3] instead proposes to leverage human judgments on relative reflectance for evaluation. However, these datasets do not provide shape or physically-based material ground truth. Synthetic datasets [59, 32, 7, 76, 74, 16] are widely used for evaluation as ground truth, as all components can be directly exported. However, these scenes are relatively simple and do not warrant generalization to complex real-world environments. Existing real object datasets [26, 19, 29, 67] are typically captured in studio setups with constrained lighting. In-the-wild object datasets, on the other hand, typically do not capture shape and ground-truth lighting, which are crucial for disentangled material evaluation through relighting. To the best of our knowledge, Lombard et al. [36, 47] is the only in-the-wild dataset providing ground-truth lighting, but is of a small scale compared to ours. In this work, we propose the first comprehensive real-world object inverse rendering benchmark with ground-truth scans and in-the-wild environment lighting.

## 3 Stanford-ORB: A Real-World Object Inverse Rendering Benchmark

Our goal is to create a quantitative evaluation benchmark for real-world 3D object inverse rendering tasks. The primary objective of object inverse rendering is to recover the 3D shape and surface material of the underlying object from images, where the input to the systems can be either a dense coverage of views of the object or as few as a single image, and optionally with additional assumptions, such as actively lighting. This task can be framed either as a per-scene optimization problem or as a learning-based one. In general, the fewer views available as input, the more challenging it is to recover accurate geometry and material, and hence the more the method relies on learned priors. In this work, we focus on evaluating the quality of the shape and material recovered by various inverse rendering methods with a comprehensive benchmark. To do this, we collected a new dataset of 14 common real-world objects with different materials, captured from multiple viewpoints in a diverse set of real-world environments, paired with their ground-truth 3D scans and environment lighting.

### 3.1 Evaluation Benchmarks

We design a number of metrics to evaluate the recovered shape and material from three perspectives. We give an outline of these benchmarks in this section and lay out the metric definitions in Section 4.

**Geometry Estimation Benchmark.** To assess the quality of the 3D shape reconstructions, we provide ground-truth 3D scans of each object. These 3D scans are obtained using a high-definition 3D scanner and carefully processed into watertight meshes. We measure the quality of the geometry estimated from different methods by comparing the predicted depth maps and normal maps to those generated from the ground-truth scans, as well as directly computing the bidirectional Chamfer Distance between the predicted meshes and the scans.

**Novel Scene Relighting Benchmark.** Evaluating surface materials is challenging as obtaining ground-truth data is generally difficult in practice, and different methods might adopt different

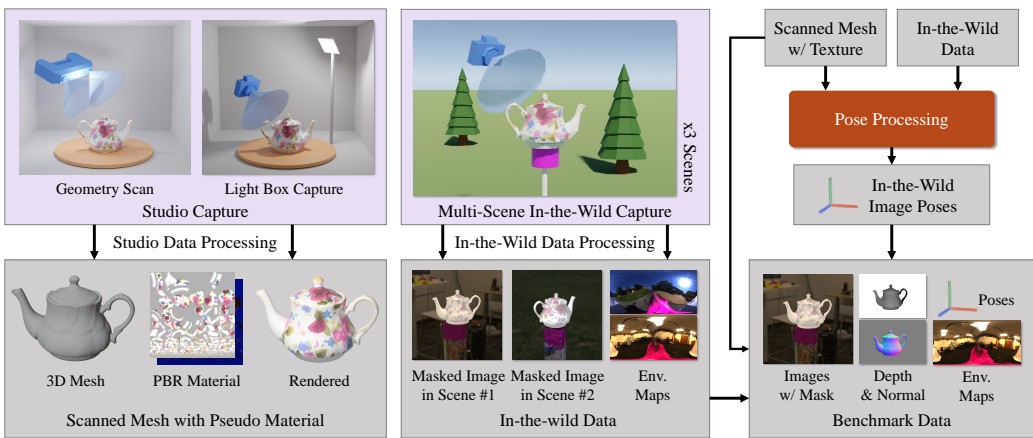

Figure 1: **Data Capture Pipeline Overview.** For each object, *Left*: we obtain its 3D shape using a 3D scanner and Physics-Based Rendering (PBR) materials using high-quality light box images. *Middle*: we also capture multi-view masked images in 3 different in-the-wild scenes, together with the ground-truth environment maps. *Right*: we carefully register the camera poses for all images using the scanned mesh and recovered materials, and prepare the data for the evaluation benchmarks. Credit to Maurice Svay [41] for the low-poly camera mesh model.

material representations. Here, we propose to evaluate the quality of material recovery through the task of relighting in a *real-world* environment unseen during training, given the *ground-truth environment lighting*, regardless of the underlying material representations. To do this, for each object, we capture multi-view image sets under multiple in-the-wild scenes, together with the ground-truth environment map without the object for each scene. To evaluate the performance of a model, we render the object using the predicted geometry and materials and the ground-truth lighting of a new scene, and compare the rendered image with the captured ground-truth image from a given viewpoint.

**Novel View Synthesis Benchmark.** In addition to the novel-scene relighting evaluation, we also perform an evaluation on the typical novel view synthesis task, where novel views of the object are rendered in the *same* scene as the input but from *novel* viewpoints, and compared to the ground-truth images. Note that this metric measures the accuracy of appearance modeling within one scene, but not necessarily the accuracy of material modeling that ensures accurate appearance in arbitrary scenes.

## 3.2 Data Capture

To support these evaluations, we capture a new real-world dataset comprised of ground-truth 3D scans, multi-view images taken in the wild, and the corresponding environment lighting. The dataset contains 14 common objects captured in 7 natural scenes. For each object, we take 60 training views and 10 testing views in high dynamic range (HDR) under 3 different scenes, resulting in a total of 42 scenes. For each testing view, we also capture an HDR panorama environment map. The ground-truth shape for each object is obtained using a professional 3D scanner. Fig. 1 gives an overview of our data capture pipeline.

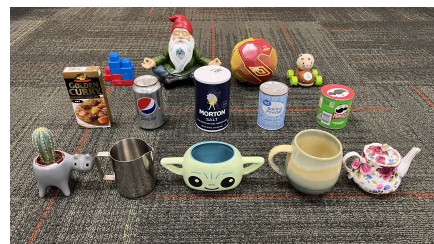

Figure 2: **Selection of Objects.** From top to bottom: *Block*, *Gnome*, *Ball*, *Car*; *Curry*, *Pepsi*, *Salt*, *Baking*, *Chips*; *Cactus*, *Pitcher*, *Grogu*, *Cup*, *Teapot*.

### 3.2.1 Object Selection

Existing work often curates a small selection of objects for evaluation specific to the focus of their proposed methods. For instance, PhySG [85] evaluates on glossy objects, and NeRD [7] tests on objects with rougher surfaces. To establish a comprehensive benchmark, we carefully select 14 objects that cover a wide variety of both geometry and material properties. Fig. 2 gives a glimpse into the list of objects. The geometry ranges from simple cylindrical shapes to complex shapes,

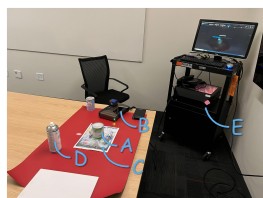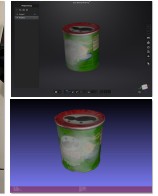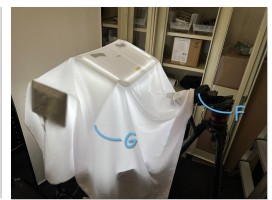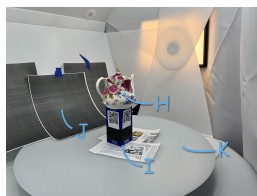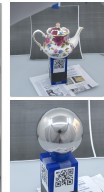

|                     |                      |
|:-------------------:|:--------------------:|
| (a) Geometry scanning | (b) Light box scanning |

Figure 3: **Studio Capture Setup.** (a) 3D shape scanning. *A*: object, *B*: hand-held EinScan Pro HD 3D Scanner, *C*: printed patterns for camera registration, *D*: spray for high-quality scanning, *E*: desktop for processing. The scanned mesh is visualized in ExScan Pro [60] and MeshLab [15] on the right. (b) Light box capture setup viewed from outside and inside. *F*: DSLR camera, *G*: cloth cover to block light from outside, *H*: object, *I*: printed patterns for camera registration, *J*: (optional) dark background for object segmentation, *K*: remote-controlled turntable. The captured image and the chrome ball are visualized on the right.

and the materials cover diffuse, glossy as well as mirror-like metal surfaces. We focus primarily on non-translucent objects in this work.

### 3.2.2 3D Shape and Appearance Capture in Studio

For each object, we obtain its ground-truth 3D shape using an EinScan Pro HD 3D Scanner. Fig. 3 illustrates the capture setup. Despite the high scanning quality in general, the scanner can struggle to reconstruct shiny and dark surfaces for some objects, and we paint these areas with a special spray designed to improve the robustness. To help with camera registration, we print geometric patterns from the HPatches dataset [1] and place them underneath the objects. Each object is scanned 2-3 times in different poses to cover the entire shape. All scans are aligned and processed in the ExScan Pro software [60] to obtain a watertight textured mesh. For computation efficiency, we reduce the number of faces in each mesh to 200,000 using the method proposed by Garland *et al.* [22] and further smooth the surfaces using the HC Laplacian smoothing algorithm from Vollmer *et al.* [68].

In addition to the 3D shapes, we also obtain pseudo material decomposition for each object by capturing them in a light box. These pseudo materials are used to refine relative camera poses for in-the-wild scenes as detailed in Section 3.3, and to build a set of high-performing baseline results on the relighting benchmark as shown in Table 2. As illustrated in Fig. 3, we place each object on a remote-controlled turntable inside a simple light box. The turntable rotates 6° at a time; a bracket of three images are taken from each view at different exposure, which are fused into a high dynamic range (HDR) image for high-quality recovery using a simplified version of the algorithm in [18]. To solve for materials, we also capture the environment map inside the light box using a 3-inch steel chrome ball, following [80]. The chrome ball is placed in the same place as the object and reflects the environment light, which allows us to recover the environment map as detailed in Section 3.3.

### 3.2.3 Image and Lighting Capture in the Wild

To build the relighting and novel view synthesis benchmarks, we capture dense multi-view images of each object in different real-world in-the-wild environments, together with the corresponding ground-truth environment lighting. Each object is captured in 3 different scenes (7 unique scenes in total), including both indoor and outdoor environments, resulting in a total of 36 captures.

Fig. 4 illustrates the capture setup. In each scene, we fixate the object to a small platform and move the camera around the object in a circle from various heights. We take images from approximately 70 viewpoints roughly uniformly covering 360° views of the objects, including 10 test views and 60 training views, which are suitable for training most existing inverse rendering methods.

For each scene, we also capture the ground-truth environment lighting for relighting evaluation. Similar to Section 3.2.2, we reuse the chrome ball and solve for the environment maps following [80]. Unlike in the studio setup, where lighting is highly constrained and static, real-world (*e.g.*,, outdoor) environments are constantly changing while we capture the data. To minimize such errors, we capture one environment map per test view, forming an 'image-envmap' pair as illustrated in Fig. 4 (b). To

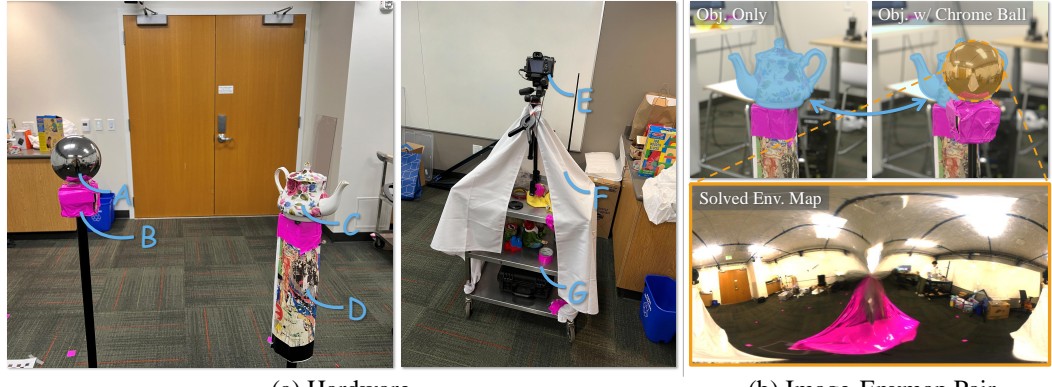

(a) Hardware        (b) Image-Envmap Pair

Figure 4: **In-the-Wild Capture Setup.** (a) Hardware for capturing. *A*: chrome ball, *B*: magenta platform for object segmentation, *C*: object, *D*: printed patterns for camera registration, *E*: DSLR camera, *F*: cloth for hiding photographer, *G*: mobile cart. (b) An example of the image-envmap pair. The environment map is solved from the reflection image on the chrome ball.

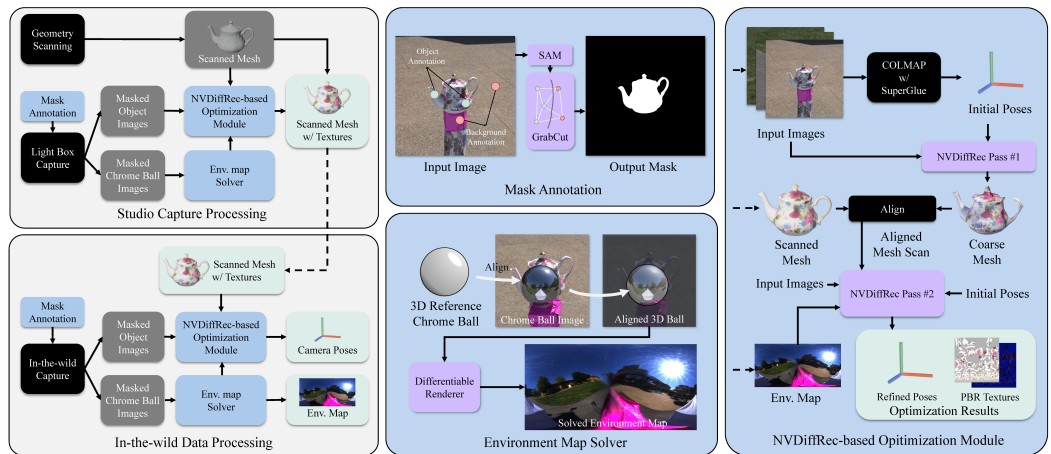

Figure 5: **Data Processing Pipeline.** *Left*: Overview of the data processing pipelines for both studio and in-the-wild captures. Three individual modules painted blue are expanded on the right. *Middle-Top*: The semi-automatically segmentation module produces object masks for all images. *Middle-Bottom*: Environment maps are solved from the chrome ball images. *Right*: Accurate camera poses are obtained from COLMAP and refined using NVDiffRec [44], given the scanned mesh and (for in-the-wild images) the pseudo materials optimized from light box captures.

avoid swapping the object for the chrome ball every time (which leads to inconsistent poses), we affix the object and only move the chrome ball in front of the object in each view. The photographer will hide themselves inside a white cloth underneath the camera to avoid being captured in the images.

For better inverse rendering quality, all images are taken in $2048 \times 2048$ resolution and in HDR by fusing multiple exposure shots. We carefully tune the exposure values for each scene, taking into account the lighting condition and the reflections on the object surface. All cameras are calibrated with a chessboard and undistorted using standard OpenCV libraries.

### 3.3 Data Processing

Next, we describe the detailed processing steps for preparing the evaluation data. In essence, we need to obtain: (1) object masks for all images, (2) environment maps solved from the chrome ball images, (3) pseudo materials for each object using the studio captures, and (4) relative camera poses across multiple views and across multiple scenes. Fig. 5 shows the overview of the pipeline, and the

details of each step are laid out below. Overall, for each object, it takes roughly 3 hours to capture the raw data (2.5 hours for the in-the-wild capture and 0.5 hour for studio capture). The data processing pipeline takes roughly 10 hours on one machine for each object, which can be paralleled, and an additional 0.5 hour of human labor to manually adjust the annotations and alignments.

**Object Masks.** Getting high-quality inverse rendering results requires highly accurate object masks. We make use of the state-of-the-art object segmentation model, SAM [27], which is trained on over a billion curated masks. Given a sparse set of around 5 query points, the model produces reasonable object masks on our images. However, we observe that the resulting masks often appear smaller than the actual object. Hence, we further refine the masks using the classic GrabCut algorithm [53].

**Environment Maps.** Given the chrome ball images, we first apply the same object masking procedure described above to segment the chrome ball. We then fit a synthetic 3D sphere to the chrome ball by optimizing its 3D location w.r.t. the camera. With a differentiable Monte-Carlo-based renderer, we optimize the environment map to fit the reflection image on the chrome ball using the single-view light estimation method proposed in [80].

**Pseudo Material Decomposition.** For the purpose of refining the in-the-wild camera poses as detailed below, we optimize a set of pseudo surface materials for each object through an inverse rendering method NVDiffRec [44], making use of the 3D scan and the images captured in the light box. Specifically, we first register all light box images using COLMAP [56, 57] and SuperGlue [55] features to obtain a set of initial camera poses. In order to fit our 3D scanned mesh to the images, we optimize a coarse mesh with NVDiffRec, and roughly align the scanned mesh to the optimized mesh manually. The ground-truth environment map is also computed from the chrome ball images. Finally, we optimize the materials with a microfacet BRDF model [17] using NVDiffRec, jointly refining the COLMAP poses like [72], given the scanned mesh and ground-truth environment map.

**Camera Pose Registration.** For both training and evaluation, we need accurate relative camera poses both between different viewpoints of a scene and across different scenes for relighting. Similarly to the registration of light box images above, we make use of the 3D scans and the pseudo materials previously solved from the studio captures. For each scene, we solve for initial poses using COLMAP and SuperGlue, optimize a coarse mesh with NVDiffRec, and align the scanned mesh to it. The camera poses are further refined using NVDiffRec given the scanned mesh, pseudo materials, and ground-truth lighting. The relative pose between each source and target scenes for relighting is thus obtained through the same canonical scanned mesh.

## 4   Benchmarking State-of-the-Art Methods

This section provides an overview of baseline methods[2], followed by evaluation details for the three benchmarks from Section 3.1 and discussion of the results.

### 4.1   Baseline Methods

Existing inverse rendering methods can be roughly categorized into

- **Material Decomposition** methods from multi-view images, such as NeRD [7], Neural-PIL [9], PhySG [85], InvRender [75], NVDiffRec [44] and NVDiffRecMC [24], which typically recovers surface materials as BRDFs and environment lighting of a scene;
- **Novel View Synthesis and 3D Reconstruction** methods from multi-view images, such as NeRF [43] and IDR [79], which focus primarily on reconstructing 3D geometry and free-viewpoint appearance synthesis of a scene, without explicit material decomposition;
- **Single-View Prediction** methods, including SIRFS [2] and SI-SVBRDF [30], which focus primarily on predicting intrinsic images, *e.g.*, depth, normal, albedo, and shading maps, and potentially full 3D reconstructions, from a single image.

Note that these methods can be either per-scene optimization-based, learning-based, or hybrid, where priors learned from large training datasets can be exploited for test time inference with limited views.

---

[2]We include several representative methods in this paper. While this is not an exhaustive list, we provide instructions for future models to submit results to the benchmark and host the most up-to-date results in our public repository: `https://github.com/StanfordORB/Stanford-ORB`.

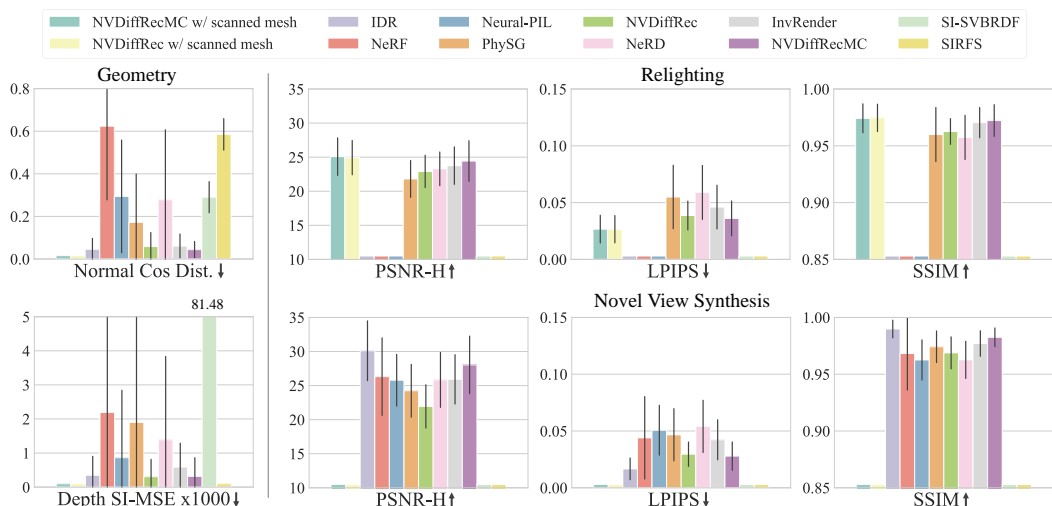

Figure 6: **Visualization of the Benchmark Comparisons.** Metrics on geometry are visualized on the left, and metrics on relighting and novel view synthesis are on the top right and the bottom right. The black line on each bar indicates the standard deviation of the scores across all scenes.

All methods assume access to images and camera poses during optimization or training. Inverse rendering methods that additionally assume access to the ground truth shape [36, 37] or ground truth lighting [48, 47, 78] are not included in this benchmark.

Furthermore, we design two strong baselines, using NVDiffRec [44] and NVDiffRecMC [24] with ground-truth 3D scans and pseudo materials optimized from light-box captures. As shown in Table 2, these two baselines achieve the best results in relighting as expected. For all methods, we use their public codebase and default configurations in all experiments, and feed in HDR images if possible by default. All models are trained and tested with $512 \times 512$ images, except SI-SVBRDF [30] which is trained and tested with $256 \times 256$ images (results are upsampled to 512 for consistency).

### 4.2 Tasks

We consider three inverse rendering tasks: geometry estimation, novel scene relighting, and novel view synthesis.

**Geometry Estimation.** We evaluate geometry reconstruction quality by comparing the rendered depth maps, normal maps, and predicted 3D meshes to the ground truth obtained from 3D scans. For per-scene multi-view optimization methods, we evaluate the predictions on held-out test views of the same scene. For single-image prediction methods, we simply run the pre-trained model [30] or optimization [2] on all test images and evaluate the results. Since neither of [30, 2] predicts novel views or full 3D shapes, we only evaluate their depth and normal predictions and omit the other two benchmarks.

To evaluate depth maps, we use the Scale-Invariant Mean Squared Error (**SI-MSE**) [21] between predicted and ground-truth images masked with ground-truth object masks: $\min_{\alpha \in \mathbb{R}} \sum_{k=1}^{K} \frac{1}{K} \|(I_{\text{gt},k} - \alpha I_{\text{pred},k}) \odot M_{\text{gt},k}\|_2^2$, where $\odot$ denotes the Hadamard product, and $\alpha$ is the scale optimized for all $K$ testing views of each scene. For normal evaluation, we compute the **Cosine Distance** between the predicted and ground-truth masked normal maps. For direct shape evaluation, we follow DeepSDF [49] to compute the bi-directional **Chamfer Distance** between $30,000$ points sampled from the predicted mesh surface and ground-truth mesh vertices. For methods that do not produce explicit meshes [79, 43, 9, 85, 7, 88, 75], we run the marching cube algorithm [38] for mesh extraction. Following IDR [79], for all methods, we take the largest connected component as the final mesh prediction. In rare cases where a method fails to converge during the training, we simply sample 30k dummy points at the origin.

**Novel Scene Relighting.** We evaluate the quality of material decomposition through the task of relighting under *novel* scenes with the ground-truth environment lighting. To do this, we take

Table 2: **Benchmark Comparison of Existing Methods**. †denotes models trained with the ground-truth 3D scans and pseudo materials optimized from light-box captures. Depth SI-MSE $\times 10^{-3}$. Shape Chamfer distance $\times 10^{-3}$.

| | Geometry | | | Novel Scene Relighting | | | | Novel View Synthesis | | | |
|---|---|---|---|---|---|---|---|---|---|---|---|
| | Depth↓ | Normal↓ | Shape↓ | PSNR-H↑ | PSNR-L↑ | SSIM↑ | LPIPS↓ | PSNR-H↑ | PSNR-L↑ | SSIM↑ | LPIPS↓ |
| NVDiffRecMC [24]† | | N/A | | **25.08** | 32.28 | 0.974 | **0.027** | | N/A | | |
| NVDiffRec [44]† | | N/A | | 24.93 | **32.42** | **0.975** | **0.027** | | N/A | | |
| IDR [79] | 0.35 | 0.05 | **0.30** | | N/A | | | **30.11** | **39.66** | **0.990** | **0.017** |
| NeRF [43] | 2.19 | 0.62 | 62.05 | | N/A | | | 26.31 | 33.59 | 0.968 | 0.044 |
| Neural-PIL [9] | 0.86 | 0.29 | 4.14 | | N/A | | | 25.79 | 33.35 | 0.963 | 0.051 |
| PhySG [85] | 1.90 | 0.17 | 9.28 | 21.81 | 28.11 | 0.960 | 0.055 | 24.24 | 32.15 | 0.974 | 0.047 |
| NVDiffRec [44] | **0.31** | 0.06 | 0.62 | 22.91 | 29.72 | 0.963 | 0.039 | 21.94 | 28.44 | 0.969 | 0.030 |
| NeRD [7] | 1.39 | 0.28 | 13.70 | 23.29 | 29.65 | 0.957 | 0.059 | 25.83 | 32.61 | 0.963 | 0.054 |
| NeRFactor [88] | 0.87 | 0.29 | 9.53 | 23.54 | 30.38 | 0.969 | 0.048 | 26.06 | 33.47 | 0.973 | 0.046 |
| InvRender [75] | 0.59 | 0.06 | 0.44 | 23.76 | 30.83 | 0.970 | 0.046 | 25.91 | 34.01 | 0.977 | 0.042 |
| NVDiffRecMC [24] | 0.32 | **0.04** | 0.51 | 24.43 | 31.60 | 0.972 | 0.036 | 28.03 | 36.40 | 0.982 | 0.028 |
| SI-SVBRDF [30] | 81.48 | 0.29 | N/A | | N/A | | | | N/A | | |
| SIRFS [2] | N/A | 0.59 | N/A | | N/A | | | | N/A | | |

decomposed results on one training scene and render them in the other 2 scenes with the respective ground-truth lighting. Note that IDR, NeRF, and Neural-PIL use implicit lighting or appearance representations and cannot directly support relighting. Hence, we do not evaluate relighting for these methods.

We adopt three widely-used metrics to compare the rendered relit images against the ground-truth images: Peak Signal-to-Noise Ratio (**PSNR**), Structural Similarity Index Measure (**SSIM**) [70], and Learned Perceptual Image Patch Similarity (**LPIPS**) [86]. Since there is an inherent scale ambiguity in material and lighting decomposition, we follow PhySG [85] and adapt all metrics to be scale-invariant by rescaling the predictions with an optimized scale for each RGB channel. Specifically, given the ground-truth image $I_{gt}$ and the predicted image $I_{pred}$, we rescale the prediction to $I'_{pred} = \{ \mathrm{argmin}_s (\|I^i_{gt} - sI^i_{pred}\|^2) \cdot I^i_{pred} \}_{i=\mathrm{R,G,B}}$ before computing the metrics.

We report PSNR in both HDR and LDR (after the standard sRGB tone mapping), denoted as PSNR-H and PSNR-L, respectively. The HDR values are clamped to a maximum of 4 to suppress the effects of over-saturated pixels (*e.g*., mirror reflection of the sun). SSIM and LPIPS are computed with LDR values. In cases where the baselines fail to converge during training, we use a constant gray image to represent the prediction.

**Novel View Synthesis.** We also report the typical novel view synthesis metrics on the *same* scene as training. Specifically, for each trained model, we render the object from test views under the same scene and compare the rendered images with ground-truth test images. The two single-view inverse rendering methods [30, 2] do not explicitly perform view synthesis, and therefore are not evaluated on this task. We use the same metrics as in relighting (without rescaling), namely PSNR-H, PSNR-L, SSIM, and LPIPS.

### 4.3 Results

We briefly summarize the key findings of the results in Table 2 and Fig. 6. For the tasks of geometry reconstruction and novel view synthesis (NVS), IDR [79] outperforms all other baselines, including NeRF [43]. A potential reason is that IDR focuses on surface reconstruction and represents the geometry using a signed distance function (SDF) [49], hence particularly advantageous in modeling these object-centric scenes. While NeRF is powerful in modeling large-scale scenes, it fails to reconstruct consistent geometry and texture details for certain scenes (see Fig. 3 in the Appendix). Inverse rendering methods that extend from IDR ( [85, 75]) or NeRF ( [9, 7, 88]) for further decomposition of lighting and surface materials tend to result in a degradation of performance in geometry reconstruction and NVS. In comparison, NVDiffRec [24] and NVDiffRecMC [24] recover better geometry, suggesting the advantages of the underlying DMTet [58]-based shape representation over SDF or NeRF. NVDiffRecMC also significantly outperforms NVDiffRec and other material decomposition baselines on NVS thanks to the differentiable Monte Carlo–based (MC) renderer.

For the task of relighting, the two most recent inverse rendering methods [75, 24] achieve better performance than the earlier ones. In particular, NVDiffRecMC again improves upon its precursor NVDiffRec and consistently outperforms all other inverse rendering baselines across all metrics, further confirming the effectiveness of the MC-based renderer. Finally, as expected, the two baselines that have access to ground truth 3D scans and pseudo-materials (marked with [†]) perform significantly better than methods without this information, and this gap suggests room for improvement for all existing methods. Here, the performance gain from using the MC renderer seems negligible.

Please refer to the Appendix for more results. In particular, we additionally evaluate and compare the albedo prediction quality of various methods using the albedo maps optimized from the studio capture data as pseudo ground-truth. The complete comparison results are summarized in Table 1 of the Appendix, and a few visual examples are provided in Fig. 3 of the Appendix.

## 5   Conclusion

We present Stanford-ORB, the first real-world evaluation benchmark for object inverse rendering tasks including shape reconstruction, object relighting, and novel view synthesis under in-the-wild environments. To build this benchmark, we capture a new dataset of 14 real-world objects under various in-the-wild scenes with ground-truth 3D scans, multi-view images, and ground-truth environment lighting. Using the dataset, we evaluate and compare a wide range of state-of-the-art inverse rendering methods. All data collected in this work are focused on daily objects and do not contain any personal information. Please refer to the website for more information on the dataset: `https://stanfordorb.github.io/`.

**Acknowledgments.**   This work was in part supported by NSF CCRI #2120095, RI #2211258, ONR MURI N00014-22-1-2740, the Stanford Institute for Human-Centered AI (HAI), Adobe, Amazon, Ford, and Google.

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

# F Additional Dataset Details

We list all the in-the-wild scenes and captured objects in Fig. 7. Our dataset is captured in 5 indoor scenes and 2 outdoor scenes (one in the daytime, one at night), and each object is captured in two indoor scenes and one outdoor scene. For each object-scene pair, we capture around 60 training images and 10 testing image-envmap pairs. The camera positions are evenly spaced in a 360-degree circle around the object, and at three different elevation levels (roughly $0°$, $20°$, and $40°$). As a result, the in-the-wild dataset consists of 2,975 object image brackets and 418 chrome ball image brackets.

For the light box capture, since the camera and the environment are fixed during the capture, we use a universal chrome ball image bracket for multiple objects captured in the same setting. There are in total two settings: For objects with white surfaces, we attach several black papers in the light box as background. For other objects, we do not add anything in the light box. In total, 872 object image brackets are captured.

All data, scripts, documents, and metadata can be accessed via `https://3dorb.github.io/`. We've made our dataset available under the MIT license. For long-term sustainability, we've housed our code on Github and our data sets on Google Drive, as they both ensure prolonged availability.

## F.1 Image Processing

We fuse the multi-exposure RAW image brackets into HDR images. In our setting, captures in each bracket share the same f-number and ISO, and only differ in the exposure times. For a bracket of raw images $R_{1,..,n}$ captured with exposure times of $t_{1,..,n}$, let $M_k$ represent the mask of non-saturated (*i.e.*, not overly exposed) pixels of each image, the output HDR image is generated by:

$$I = \frac{\sum_{i=1}^{n} R_i \odot M_i t_i}{\sum_{i=1}^{n} M_i t_i}, \tag{1}$$

where $\odot$ represents the Hadamard product.

We also provide LDR images for the following processing steps and the final benchmark. As some of the HDR images are too bright, we rescale them to ensure that $99\%$ of the rescaled pixel values are less than 1. The scale factor is calculated globally across all images for each object in each scene to keep the brightness of the images consistent. The scaled images are then tone-mapped with the commonly used sRGB formula [62].

For the camera calibration, we directly extract the focal length ($f_x, f_y$) from the EXIF metadata, and optimize the principle point ($c_x, c_y$) and distortion parameters (2 parameters for radial distortion $k_1, k_2$ and 2 for tangential distortion $p_1, p_2$). To do that, we print the chessboard pattern on a paper sheet, as shown in Fig. 8 on the right. We then take pictures from multiple viewpoints and apply the calibration algorithm in the OpenCV library to solve for the camera intrinsics. All images in the dataset are then undistorted using the estimated camera intrinsics.

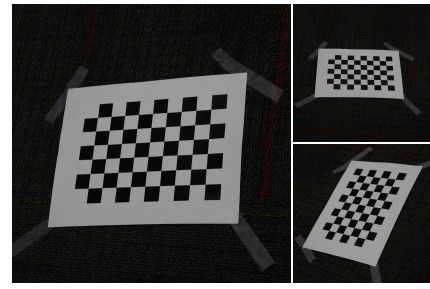

Figure 8: Example images of the printed chessboard patterns for calibration.

## F.2 Environment Map Solver

As discussed in the paper, we fit a synthetic 3D ball to recover the environment map from the chrome ball image. This process consists of two steps. The first step optimizes the pose of the synthetic ball w.r.t. the camera, by minimizing two losses given the masks of the chrome ball and the rendered synthetic ball: $\mathcal{L}_2$ loss between the masks of two balls, and the $\mathcal{L}_2$ loss between the center pixels of two balls. The second step optimizes the environment map from the synthetic ball. Here in each iteration, we randomly sample a batch of rays from the camera and render them with a differentiable path tracer. The environment map is optimized from the $\mathcal{L}_2$ loss between the rendered color and the ground truth color from the image.

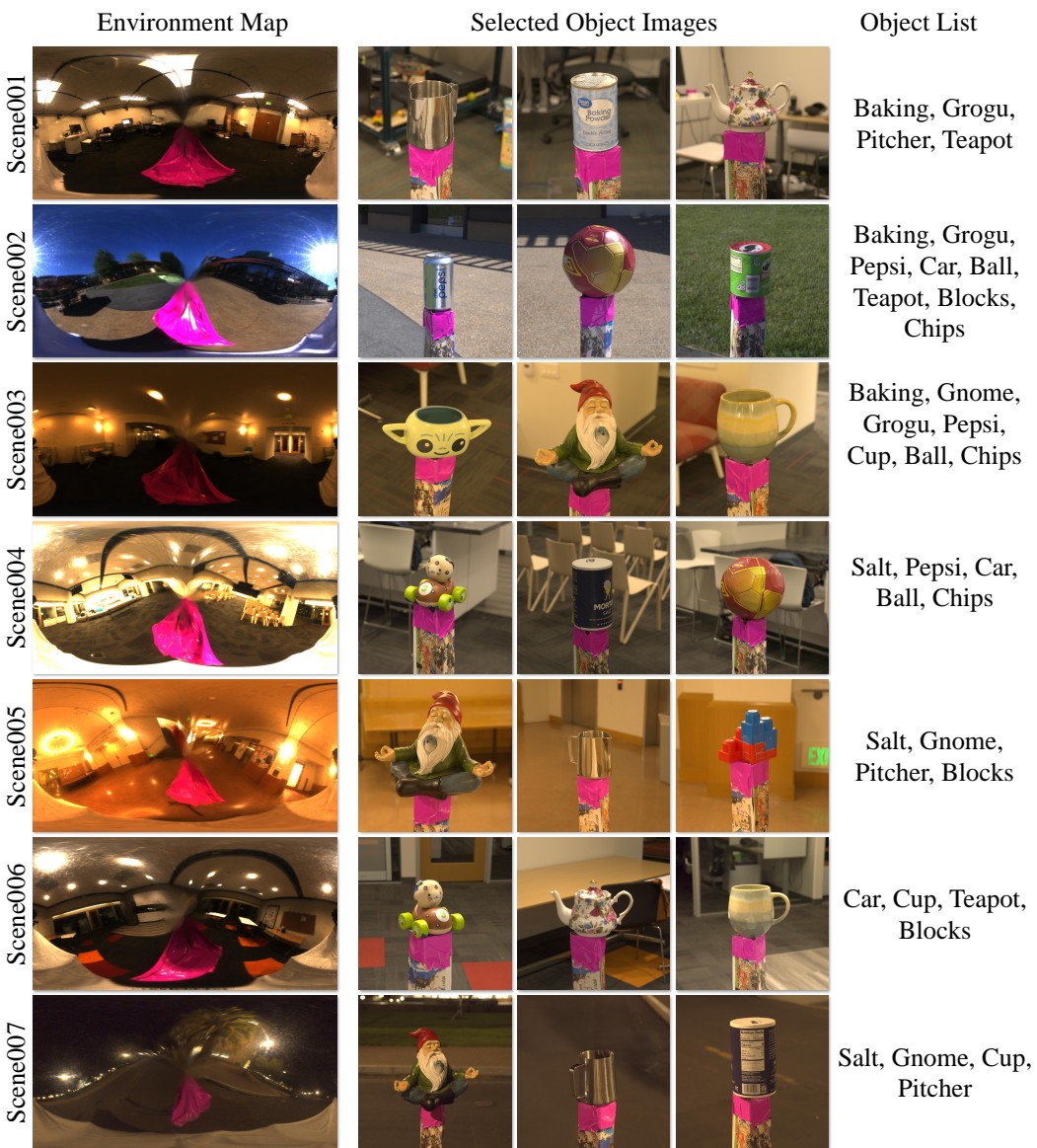

Figure 7: Overview of the dataset. We capture the objects in 7 different in-the-wild scenes. For each scene, we showcase the environment maps on the left, three examples of the objects captured in the middle, and the full list of objects in the last column.

### F.3 Material Optimization for Studio Data

To process the studio data, the pseudo surface materials of the captured object are generated from an NVDiffRec-based [44] optimization pipeline. As explained in the paper, it is done with two optimization steps, one to obtain a coarse mesh from scratch and align it with the scanned mesh, and the other to optimize the materials from the aligned scanned mesh. Here we provide additional details of the two passes. In the first pass, we train the DMTet module of the NVDiffRec from scratch for 400 iterations with a batch size of 4 and a learning rate of $3 \times 10^{-2}$. The coarse mesh is then extracted from the resulting SDF using Deep Marching Tetrahedra [58]. After aligning the optimized mesh to the scanned mesh, the second pass optimizes the DLMesh module of NVDiffRec for 2000 iterations, given the scanned mesh and the environment map from the chrome ball. Here, all trainable components are frozen except the material textures and camera poses. The learning rate is set to $3 \times 10^{-3}$ and the batch size is set to 16. We use the same losses proposed in NVDiffRec.

### F.4 Pose Optimization for In-the-wild Data

To process the in-the-wild data, the same NVDiffRec-based optimization module is deployed to obtain accurate camera poses of in-the-wild images. The setting of this module resembles the previous one, with only a few differences in the second pass. Here, the material textures are frozen, and the environment map and the camera poses are optimized. The training consists of 2000 iterations with a batch size of 16. In addition to the losses in NVDiffRec, we also apply the regularization loss from NeRF$--$[72] to regularize the camera transformations. For each data point, we try multiple runs with different loss weights and learning rates, and select the set of parameters for the best performance in terms of PSNR on the test images.

## G   More Results

As a supplement to Table 2 in the main paper, Table 3 presents the full benchmark results with all metrics and baselines, including the standard deviations.

**Albedo Benchmark.**   In addition to the three benchmarks proposed in the main paper, we also compare albedo maps predicted by baselines with the pseudo ground truth albedo maps optimized from the studio capture. We compute scale-invariant metrics, PSNR-L, SSIM, and LPIPS, as introduced in Section 4.2 of the main paper to evaluate the results. The results are reported in Table 3. We do not evaluate other material properties (such as roughness) as different methods adopt different material representations (*e.g.*,, Lambertian, Phong, microfacet BRDFs).

**More Qualitative Results.**   Finally, we show qualitative results of all baselines on two example scenes in Fig. 9, including the normal maps, depth maps, view synthesis images, relighting images, and albedo maps. In particular, to examine why IDR [79] achieves a higher performance in the geometry reconstruction and novel view synthesis benchmark, Fig. 10 provides a visual comparison of the two methods on two example scenes. IDR reconstructs sharper texture details compared to NeRF, yielding a higher novel view synthesis score.

We also observed that despite having similar geometry, objects with different appearances can lead to drastically different inverse rendering results. For example, although the *Baking* can and the *Salt* can have similar cylindrical shapes, NVDiffRecMC [24] achieves a PSNR-HDR score of 31.0 (view synthesis) / 24.4 (relighting) on *Baking* and 17.9 (view synthesis) / 7.5 (relighting) on *Salt*. Visual comparisons are presented in Fig. 11. In particular, the black texture of the *Salt* can poses significant challenges to the inverse rendering task due to the inherent ambiguity between lighting and texture.

## H   Limitations

Although the current version of the dataset already contains a variety of objects, it 300 does not cover translucent objects, like glass. It also remains challenging to capture thin, deformable objects like leaves as their shape will change while being transported to different scenes, breaking the consistency for novel scene relighting, and the scanner also tends to fail in capturing extremely intricate geometry details. We plan to continue expanding the collection of objects. Currently, we only capture one object in a scene at a time. Extending it to multi-object scenes will allow us to further analyze the

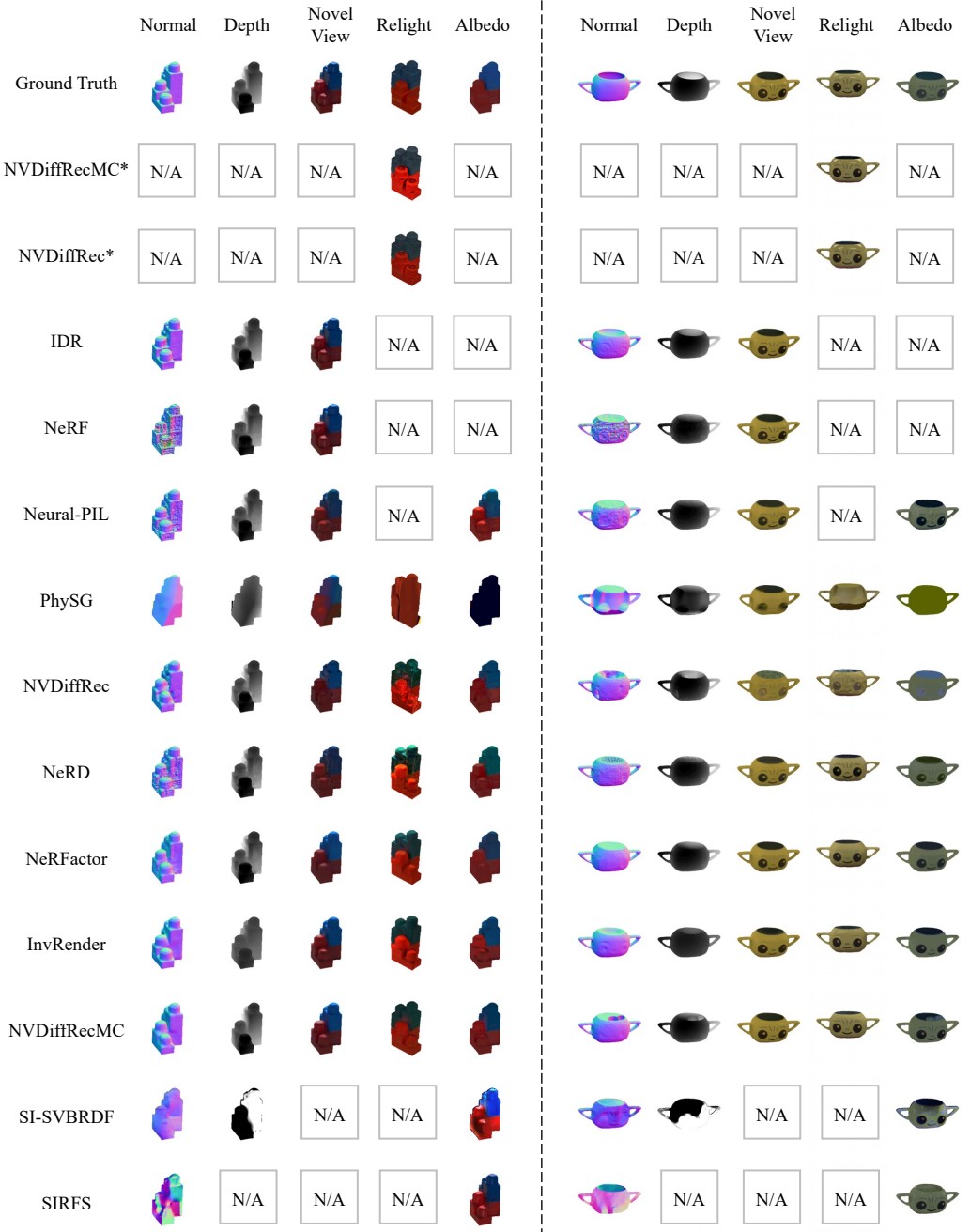

Figure 9: **Qualitative Comparisons of Baseline Methods**. We show qualitative results on two example scenes. * denotes models trained with the ground-truth 3D scans and pseudo materials optimized from light-box captures.

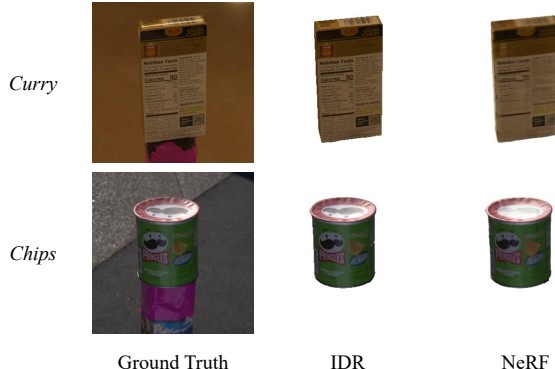

| Ground Truth | IDR | NeRF |

Figure 10: **Comparisons of IDR and NeRF on Novel View Synthesis.** We compare the novel view synthesis results of IDR and NeRF on two example scenes, *Chips* and *Curry*. IDR recovers more texture details and hence achieves higher scores.

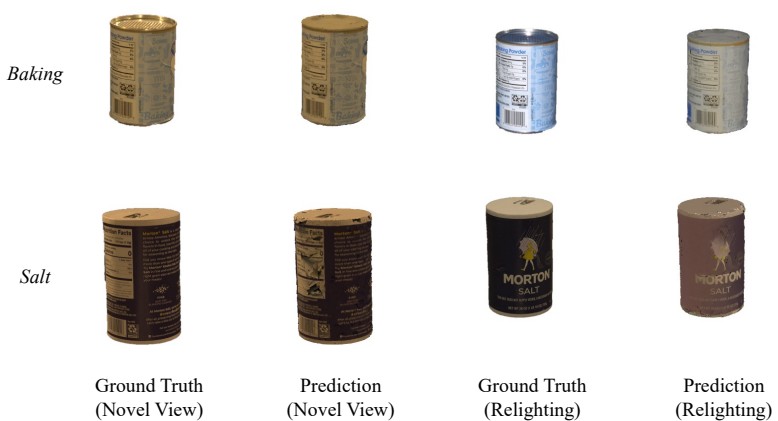

| Ground Truth (Novel View) | Prediction (Novel View) | Ground Truth (Relighting) | Prediction (Relighting) |

Figure 11: **Result Comparisons on Two Can-shaped Objects with Different Appearances.** We compare the novel view synthesis and relighting results of NVDiffRecMC [24] on two objects: *Baking* and *Salt*. Despite their similar geometry, these two objects lead to drastically different results due to the different in their textures.

effects of global illumination. The current environment maps extracted from individual chrome ball images do not capture the environment behind the ball, and we plan to align and stitch the maps from all views, which will enable lighting evaluation.

# I  Societal Impact

The purpose of this work is to facilitate the development of general object inverse rendering methods by contributing a real-world evaluation dataset and benchmark. We believe there is no direct negative societal impact resulting from this benchmark. The technology of inverse rendering has numerous applications that will have positive societal impact, such as improving the realism of computer graphics for virtual and augmented reality, or aiding in scene understanding for robotics applications. However, if leveraged for malicious intentions, it could create a significant negative impact on society. For instance, extremists might exploit an inverse rendering methods to generate fake imagery, commonly known as 'deepfakes', which may lead to the spread of misinformation or even intimidate the public. Nevertheless, we believe these possibilities can be eliminated with sufficient supervision from the governments and the research communities. It is important to note that our work focuses solely on common everyday objects, steering clear of any personal information or hazardous items.

Table 3: **Full Benchmark Comparison of Existing Methods.** †denotes models trained with the ground-truth 3D scans and pseudo materials optimized from light-box captures. Depth SI-MSE $\times 10^{-3}$. Shape Chamfer distance $\times 10^{-3}$.

| | Geometry | | | Novel Scene Relighting | | | | Novel View Synthesis | | | | Albedo | | |
|---|---|---|---|---|---|---|---|---|---|---|---|---|---|---|
| | Depth↓ | Normal↓ | Shape↓ | PSNR-H↑ | PSNR-L↑ | SSIM↑ | LPIPS↓ | PSNR-H↑ | PSNR-L↑ | SSIM↑ | LPIPS↓ | PSNR-L↑ | SSIM↑ | LPIPS↓ |
| NVDiffRecMC [24]† | N/A | N/A | N/A | 25.08±2.72 | 32.28±2.48 | 0.974±0.013 | 0.027±0.012 | N/A | N/A | N/A | N/A | N/A | N/A | N/A |
| NVDiffRec [44]† | N/A | N/A | N/A | 24.93±2.48 | 32.42±2.35 | 0.975±0.012 | 0.027±0.012 | N/A | N/A | N/A | N/A | N/A | N/A | N/A |
| IDR [79] | 0.35±0.55 | 0.05±0.05 | 0.30±0.35 | N/A | N/A | N/A | N/A | 30.11±4.34 | 39.66±2.82 | 0.990±0.008 | 0.017±0.009 | N/A | N/A | N/A |
| NeRF [43] | 2.19±4.27 | 0.62±0.34 | 62.05±179.10 | N/A | N/A | N/A | N/A | 26.31±5.63 | 33.59±7.28 | 0.968±0.032 | 0.044±0.036 | N/A | N/A | N/A |
| Neural-PIL [9] | 0.86±1.96 | 0.29±0.26 | 4.14±16.09 | 21.81±2.68 | 28.11±3.15 | 0.960±0.024 | 0.055±0.028 | 25.79±3.75 | 33.35±3.63 | 0.963±0.017 | 0.051±0.022 | 36.89±3.76 | 0.970±0.014 | 0.058±0.023 |
| PhySG [85] | 1.90±5.47 | 0.17±0.22 | 9.28±20.50 | 22.91±2.35 | 29.72±2.28 | 0.963±0.011 | 0.039±0.013 | 24.24±3.84 | 32.15±4.44 | 0.974±0.014 | 0.047±0.023 | 36.82±4.32 | 0.960±0.039 | 0.065±0.041 |
| NVDiffRec [44] | 0.31±0.50 | 0.06±0.07 | 0.62±0.74 | N/A | N/A | N/A | N/A | 21.94±3.14 | 28.44±3.54 | 0.969±0.014 | 0.030±0.011 | 40.70±4.14 | 0.983±0.011 | 0.043±0.021 |
| NeRD [7] | 1.39±2.42 | 0.28±0.32 | 13.70±30.28 | 23.29±2.44 | 29.65±2.16 | 0.957±0.019 | 0.059±0.023 | 25.83±3.99 | 32.61±3.92 | 0.963±0.016 | 0.054±0.023 | 40.59±3.51 | 0.981±0.012 | 0.058±0.026 |
| NeRFactor [88] | 0.87±1.23 | 0.29±0.41 | 9.53±21.43 | 23.54±3.15 | 30.38±3.08 | 0.969±0.013 | 0.048±0.019 | 26.06±4.22 | 33.47±4.48 | 0.973±0.014 | 0.046±0.020 | 40.11±3.82 | 0.985±0.007 | 0.051±0.020 |
| InvRender [75] | 0.59±0.69 | 0.06±0.06 | 0.44±0.55 | 23.76±2.72 | 30.83±2.57 | 0.970±0.013 | 0.046±0.019 | 25.91±3.58 | 34.01±3.04 | 0.977±0.011 | 0.042±0.017 | 40.32±3.85 | 0.982±0.010 | 0.049±0.020 |
| NVDiffRecMC [24] | 0.32±0.54 | 0.04±0.04 | 0.51±0.57 | 24.43±2.95 | 31.60±3.15 | 0.972±0.014 | 0.036±0.015 | 28.03±4.18 | 36.40±3.47 | 0.982±0.008 | 0.028±0.012 | 41.60±3.87 | 0.986±0.009 | 0.038±0.018 |
| SI-SVBRDF [30] | 81.48±17.85 | 0.29±0.07 | N/A | N/A | N/A | N/A | N/A | N/A | N/A | N/A | N/A | 34.94±3.51 | 0.957±0.021 | 0.064±0.023 |
| SIRFS [2] | N/A | 0.59±0.07 | N/A | N/A | N/A | N/A | N/A | N/A | N/A | N/A | N/A | 37.90±3.01 | 0.978±0.012 | 0.039±0.018 |

