# OpenReview forum: "Stanford-ORB: A Real-World 3D Object Inverse Rendering Benchmark"
_NeurIPS.cc/2023/Track/Datasets_and_Benchmarks — NeurIPS 2023 Datasets and Benchmarks Poster_

### Official Review · Reviewer_8Dpe · 2023-07-20
**Review for "Real-World 3D Object Inverse Rendering Benchmark"**

**Rating:** 6
**Confidence:** 5

**Strengths:**

The work has significance in overcoming the limitations of existing inverse rendering datasets, that, if real-world, lack detailed object-centric geometry focusing solely on lighting and appearance disambiguation, or when providing richer detail, are synthetically generated.

The authors have invested a significant amount of effort in the incredibly cumbersome data acquisition process that goes into full 360 scene understanding of a single scene, having acquired dense full-range 3D scans, high-dynamic range environment maps, and multi-viewpoint images. The data acquisition process is well detailed, both in the main paper and in the supplementary material.

The challenge addressed of acquiring full-scale real-world 3D scene data for tasks in inverse rendering has the benefit of enabling the research community to stress test multiple aspects of inverse rendering algorithms, such as geometry, illumination, and appearance, with respect to real-world ground truth.

**Additional Feedback:**

N/A

**Clarity:**

The paper as written is easy to follow. However, as discussed before, there are details missing that should be incorporated to strengthen the paper.

Minor:

"envmap" should be pre-expanded to "environment map" prior to its occurrence in the text (line 192)

Line 194: fixate --> perhaps "affix" or "anchor"

**Correctness:**

Lines 121-122: The paper claims that extracting explicit shapes from the implicit neural representations (INRs) used by many inverse rendering methods is still an open question, and therefore avoid estimating the Chamfer distance. In the case of IDR, this claim is technically incorrect, as IDR outputs a signed distance field (SDF) on which the authors use Marching Cubes to extract a 3D surface and compute Chamfer distance. Though NeRF does not output the SDF, the authors of NeuS (Wang et al., 2021) compare their proposed approach of learning a volume density to SDF transformation to a simple baseline of thresholding the volume density to get an occupancy value, which can also be converted to a 3D mesh using Marching Cubes. (The quality of a thresholded mesh is obviously inferior to an approach geared toward predicting the SDF, but it can be done.) The paper could benefit from estimating the actual (IDR) and approximate (NeRF) mesh and measuring the Chamfer distance, as a full-fledged insight of the quality of a 3D surface reconstruction from an inverse rendering method could be beneficial for use cases looking to take advantage of the 3D surface for purposes other than rendering.

Wang, P., Liu, L., Liu, Y., Theobalt, C., Komura, T., & Wang, W. (2021). NeuS: Learning neural implicit surfaces by volume rendering for multi-view reconstruction. NeurIPS 2021.

**Documentation:**

The submission needs completeness to enable Reproducibility (as defined in https://arxiv.org/pdf/2003.12206.pdf, taken from the NeurIPS 2023 D&B page). Documentation on how to use the code is sparse, consisting of a README with a sentence on how to run an approach, but with no guidance on what outputs to expect---at command prompt or in a file, for instance, and if in a file, how various metrics will be arranged. With full disclosure, this reviewer has not run the code, however, when examining the code, it was difficult to locate the format in which outputs would be generated. Metrics appear to be computed in image/utils/test.py, and outputs may be stored in logs/leaderboard with abbreviations corresponding to the scores discussed in the paper. Details on these points should be included in the README.

**Ethics:**

No ethical concerns have been identified.

**Limitations:**

Though the paper contains a coverage of limitations, it lacks a discussion of the potential negative social impact of their work.

Within the limitations, the claim that the dataset contains a variety of objects is not well-substantiated, as indicated above. It would be more beneficial to frame the discussion of limitations from the standpoint of the current narrow range of objects, and how should future collections be structured to diversify the range.

There are concrete negative social impacts of 'unchecked' inverse rendering that should be discussed in the paper. High-quality inverse rendering methods pave the way for high-fidelity 3D image manipulation, where traditional techniques to detect image forgery (e.g., Kee et al.) are likely to fail. 3D image manipulation has been an area of significant concern when it has come to the manipulation of faces, however, manipulation of objects can have negative consequences ranging from socially concerning to devastating, with legal and national security implications. E.g., imagine being able to insert a nuclear missile into an image with high fidelity (even though at least one prior attempt was successfully detected: https://www.npr.org/2008/07/11/92442928/photo-of-irans-missile-launch-was-manipulated), or a firearm in the hands of an innocent person in a photograph.

[It is understandable that the paper is a benchmark for other algorithms rather than an algorithm itself. However, the very contribution of a benchmark serves as the channel for future researchers to contribute more improved algorithms, making it imperative for benchmark methods to share in the burden of negative consequences.]

Kee, E., O'Brien, J. F., & Farid, H. (2014). Exposing Photo Manipulation from Shading and Shadows. ACM Trans. Graph., 33(5), 165-1.

**Opportunities For Improvement:**

As a benchmark work, the paper has severe shortcomings in an understanding of the rationale for choosing the particular selection of objects, and in the evaluation procedure itself.

As far as the object selection goes, it is not immediately clear why the set of objects selected was restricted to 12. If the reason is because of the difficulty of conducting data acquisition, then, in order to be a benchmark, it is imperative that the selected object (and similarly scene set) is carefully performed so that a small set of objects has the potential to span a wide enough range of cases that current and future inverse rendering algorithms are expected to encounter. However, it is not immediately clear that the objects show diversity. For instance, 3 of the objects have the same overall geometry (cylindrical), a fourth is close to cylindrical. Two of the cylindrical objects appear to present the same challenge in terms of information content (i.e., textured side exterior, non-textured upper surface). While it's not a concern that the size is tabletop as such, it is a concern that all objects appear to occupy a narrow aspect ratio range (i.e., nearly 1:1). Is there a reason why an object of a different type of geometry, e.g., a complex object such as a small potted plant, a flat object such as a smartphone, or an object such as a shoe were not chosen? A discussion of the design choices for the object selection is critical to strengthen the paper as a benchmark.

The paper lacks an explanation and discussion of the results obtained and their significance. The only report of results consists of the quantitative values obtained in Table 2 and Figure 6 in the main paper, quantitative values in Tables 1 and 2 in the supplementary, and qualitative visuals in Figure 3. Figure 6 in the main paper appears to be a restatement of Table 2 with standard deviation, there is no need to have both, as without an explanation, apart from the standard deviation, the distinction between them is unclear. The table can be augmented with the standard deviation. There is no discussion of the reasons for differences in outputs generated by the evaluated approaches. Lines 289-290 do not shed light on why it is surprising that IDR outperforms NeRF for the dataset provided. The paper needs to be strengthened with a comprehensive discussion of potential reasons for success or failure of various approaches. For instance, information should be provided on success or failure anchored on the geometry of the object and/or the lighting environment.

For inspiration on detailed benchmark analyses, an inspection of benchmark papers such as Dumitrescu et al. (2021) could serve beneficial.

Dumitrescu, S. D., Rebeja, P., Lorincz, B., Gaman, M., Avram, A., Ilie, M., ... & Patraucean, V. (2021, June). Liro: Benchmark and leaderboard for romanian language tasks. In Thirty-fifth Conference on Neural Information Processing Systems Datasets and Benchmarks Track (Round 1).

Is the total number of training images 60 viewpoints per capture * 36 captures or 2,160? Similarly is the total number of test images 360? Given the limited diversity of the backgrounds (only 7 scenes total), what is the extent to which overfitting occurs with the use of 2,160 images? It is unclear whether the evaluations used pre-trained networks and retrained them, or trained the models from scratch.

Following the above, are summary statistics provided over the entire list of 2,160 images? There may need to be breakdowns per object and per scene.

**Relation To Prior Work:**

Distinctions from the prior work are discussed. However, it is not clear why a geometry-only dataset such as Objaverse is discussed in the table. It does not serve any purpose -- there are a wide range of geometry-only datasets, and selecting Objaverse appears to be arbitrary. It is recommended to remove that and similar datasets from the table.

**Summary And Contributions:**

The paper provides a new benchmark for real-world 3D object-centric inverse rendering, consisting of ground truth 3D scans, environment maps, and multi-viewpoint images, captured for 12 real-world objects placed in 3 real-world scenes per objects (spread over 7 real-world total scenes). The paper also generates metrics for various inverse rendering algorithms spanning rendering while retrieving geometric structure, new scene relighting, and novel view synthesis.

---

> ### Author Response · Authors · 2023-08-21
> **Response for reviewer 8dpe's review**
>
> **Q1: The variety of objects is limited**
>
> We carefully selected a set of objects covering various combinations of different geometries (ranging from simple shapes _e.g._ cylinders, spheres and cubes to more complex ones _e.g._ teapots and toys) and surface materials (including shiny plastic and ceramics _e.g._ mugs, metal _e.g._ pitcher and coke can, as well as diffuse _e.g._ salt can).
> Note that while some objects appear to have similar shapes, such as `Salt` and `Baking`, they have very different textures, which brings diversity to the benchmarking setup since the inverse rendering methods are designed to solve both geometry and material. In practice, the texture difference indeed leads to drastically different inverse rendering performance.
> For instance, on the brighter `Baking` can, NVDiffRecMC achieves PSNR-HDR scores of 31.0 (view synthesis) / 24.4 (relighting), whereas on the `Salt` can, the scores drop drastically to 17.9 (view synthesis) / 7.5 (relighting), due to the ambiguity of lighting and texture brightness.
> We have included a visual example showcasing this difference in the supplementary material.
> We believe that including these different instances provides a comprehensive benchmarking analysis from various perspectives.
>
> We are continuing to expand object selection and the dataset. Thanks to the suggestions from reviewers `uqj7` and `8Dpe`, we have added to the dataset two new objects captured in both studio setup and 6 “in-the-wild” scenes, 3 for each object: (1) `cactus` (a small potted plant), (2) `curry` (a thin paper box), which further adds to the diversity of the selection. The new collection of objects is visualized in Figure 2 of the revised paper. We have also updated all the evaluation results incorporating these new scenes.
>
> Moreover, we would like to point out that many existing inverse rendering methods are computationally expensive, and that maintaining a reasonable dataset size is crucial to make the benchmark practically useful. For instance, NeRD requires per-scene optimization, and each scene takes 21 hours on one GPU, summing up to a total of 882 GPU hours on the entire benchmark of 42 scenes. We think the current version of the dataset is already suitably sized, ensuring reasonable computation efficiency for benchmark evaluation.
>
> Finally, we agree with suggestions from reviewers `DSB3`, and plan to create a set of detailed instructions for capturing the data which will allow other researchers to contribute to the dataset.
> The hardware involved in the capturing process is pretty standard and lightweight.
> We will also be actively maintaining and expanding the benchmark, and would be more than happy to work closely with other researchers who are interested in contributing to the dataset.
>
>
> **Q2: Result discussions, in particular why IDR is better than NeRF**
>
> Thanks for bringing this up. We have examined the results more carefully and included a discussion of interesting findings in the revised paper (marked in blue). Here, we included a summary of the key takeaways:
>
> - The performance on the geometry and view synthesis benchmark are highly correlated.
> - We observed that IDR obtained better performance on the geometry and view synthesis benchmark compared to NeRF. While NeRF is powerful for fitting complex objects and achieves superior performance in scene reconstruction, it suffers from lower surface reconstruction fidelity compared to SDF-based methods (_e.g._ IDR) which is particularly advantageous in object-centric reconstruction.
> - For the relighting benchmark, NVDiffRecMC achieves the best performance among all methods, including significantly outperforming NVDiffRec, which evidently shows the effectiveness of its differentiable Monte-Carlo-based renderer.
>
> More discussions are included in Section 4.1 of the revised paper.
>
>
> **Q3: Are the baselines trained from scratch or pre-trained?**
>
> All baselines except SI-SVBRDF [36] are optimization-based and are trained from scratch for each scene. For SI-SVBRDF [36] which is a learning-based method, we simply use the publicly released pre-trained model to run inference on the test images.
>
>
> **Q4: Mesh-based geometry evaluation**
>
> Thanks for the suggestion. We have included numerical evaluation on explicit 3D shapes using the bidirectional Chamfer distance between predicted (extracted) meshes and the ground-truth 3D scans. For methods using an implicit field representation, we apply the standard marching cube algorithm to extract explicit meshes. We follow the implementation from DeepSDF and sample 30000 points from the predicted mesh surface. Results have been included in Table 2 in the revised paper.
>
> We observed that the shape error in general correlates with the depth and normal errors, as expected. IDR obtained lower shape errors compared to NeRF, partially due to larger errors introduced in the marching cube procedure for the latter.

---

> > ### Author Response · Authors · 2023-08-21
> > **Response for reviewer 8dpe's review (cont.)**
> >
> > **Q5: Total number of images**
> >
> > We set our goal to capture 60 training views and 10 testing views per object per scene.
> > This would result in a total of (60+10) * 12 * 3 = 2,520 images (excluding the newly captured two objects in the rebuttal).
> > In practice, we ended up with a total number of 2,561 (2,975 with the two new objects) images as reported in the supplementary material due to various reasons, including duplicate captures, bad imaging (eg, blur, out-of-focus) and pose registration failure from COLMAP.
> > We will release all the data and a detailed statistics summary on the website (including the two new objects).

---

> ### Author Response · Authors · 2023-08-28
> **Further questions before the discussion ends?**
>
> We hope our responses have addressed all of the concerns the reviewer has. As the discussion window closes in a day, if there are any more questions, we would be more than happy to clarify before then. We would also love to hear it if the responses have changed the reviewer's opinion on our submission. Thanks again for the valuable feedback!

---

> > ### Comment · Reviewer_8Dpe · 2023-08-29
> > **Concerns addressed**
> >
> > Generally speaking, most of my concerns are addressed. I will update my score.
> >
> > Just a cautionary note: in two places in the revised paper, the word 'significantly' (lines 306 and 308). It is recommended to replace with a different word, as the term is technical, and when used in scientific writing, must be followed with the outcomes of a statistical significance test. For 300, a qualifier is unnecessary. 306 --> it is okay to say '...baselines outperform methods that lack the information...'

---

> > > ### Author Response · Authors · 2023-08-29
> > > **Will update the text**
> > >
> > > Thanks for the comment. We are glad that the concerns have been addressed. We will carefully refine the texts in the final version.

---

### Official Review · Reviewer_DSB3 · 2023-07-21
**Comprehensive dataset for quantitative inverse rendering evaluation**

**Rating:** 7
**Confidence:** 4
**Clarity:** The paper is easy to read.

**Strengths:**

- The dataset is comprehensive and includes ground truth for geometry, and images for relighting and novel view synthesis.

- Data is captured in real environments. This is in contrast to artificial light stage setups which simplify acquisition but produce data captured in an unrealistic environment.

- The paper evaluates multiple state-of-the-art methods.

**Additional Feedback:**

The presented dataset is important to reduce the sim-to-real gap that currently exists for the evaluation of inverse rendering methods.
An experimental comparison highlighting differences with the evaluation on synthetic data would have made the contribution even stronger.

**Correctness:**

Dataset construction seems sound.
The position and size of the chrome ball are not optimal and are a tradeoff to make the data capture feasible.
Is the error introduced by this known?

**Documentation:**

Documentation of the dataset seems sufficient. Raw data and processing scripts are not released yet.

**Ethics:**

The dataset contains no personal data. No additional review needed.

**Limitations:**

I could not find the discussion about negative societal impacts in the supplement but the datasheet shows that a negative impact is unlikely.

**Opportunities For Improvement:**

- Creating synthetic test data is significantly easier than creating a real-world test set. Experiments with synthetic data as a comparison could be used to show how significant the sim-to-real gap is.

- While the paper reports quantitative results for some state-of-the-art methods, the discussion is very short and there are almost no visual examples shown of the output of the methods. Highlighting potential problems that current methods have and showing examples of reconstructions can be helpful to better understand the results and derive future directions of research for this task.

- The motivation for not extracting an explicit shape in l. 120-122 is strange because generating explicit depth values by rendering a depth map seems to be not a problem.

**Relation To Prior Work:**

Related work is adequate but consider adding

Rudnev et al., "NeRF for Outdoor Scene Relighting" ECCV 2022.

**Summary And Contributions:**

The paper presents a novel dataset for the quantitative evaluation of inverse rendering methods.
The dataset contains images of objects taken in multiple illuminations that allow quantifying the relighting quality of inverse rendering methods across environments.
In addition, the dataset provides geometry ground truth and the paper shows an evaluation of state-of-the-art methods.

---

> ### Author Response · Authors · 2023-08-21
> **Response for reviewer dsb3's review**
>
> **Q1: Comparison with synthetic datasets**
>
> The intention of this real-world benchmark is not to replace synthetic benchmarks, and we believe the latter has great potential as rendering engines continue to improve. Rather, this benchmark is designed to compliment synthetic benchmarks, with a strong focus on sophisticated and challenging real-world scenarios.
>
> We highlight two characteristics of our real-world dataset that are not present in synthetic ones.
> 1. *Inclusion of natural capturing noises.* Our captured scenes consist of various combinations of indoor and outdoor, daytime and nighttime environments and well represent the natural capturing noises that one would typically encounter in real-world applications. It is more difficult for a synthetic rendering engine to reproduce these real-world challenges than simply capturing them.
>
> 2. *Inclusion of real-world reflection effects.* For some of the training data with highly shiny objects, the reflection of the camera and the cart (shown in Figure 4)  are also captured in the image. In some views, the camera also casts shadow on the object and changes the illumination effects. These shading effects can hardly be shown in synthetic datasets, and are definitely worth taking into account in a benchmark as they pose additional challenges for the baselines.
>
> In this work, we have made a significant effort in evaluating a wide range of representative existing methods on our benchmark.
> Despite that, unfortunately, we believe it is extremely difficult to establish a comprehensive side-by-side comparison between our benchmark and the existing synthetic benchmarks, as the objects and scenes are very different, and different methods opt to evaluate on different datasets.
> Nevertheless, we hope having this standard benchmark available will be a good starting point and that valuable insights will continue to emerge as more and more methods begin to evaluate on the same benchmarks.
>
>
> **Q2: Discussion of the results and visual results**
>
> Thanks. We have examined the results more carefully and included a discussion of interesting findings in the revised paper (marked in blue). Here, we included a summary of the key takeaways:
>
> - The performance on the geometry and view synthesis benchmark are highly correlated.
> - We observed that IDR obtained better performance on the geometry and view synthesis benchmark compared to NeRF. While NeRF is powerful for fitting complex objects and achieves superior performance in scene reconstruction, it suffers from lower surface reconstruction fidelity compared to SDF-based methods (_e.g._ IDR) which is particularly advantageous in object-centric reconstruction.
> - For the relighting benchmark, NVDiffRecMC achieves the best performance among all methods, including significantly outperforming NVDiffRec, which evidently shows the effectiveness of its differentiable Monte-Carlo-based renderer.
>
> More discussions are included in Section 4.1 of the revised paper.
>
> Due to the page limit, we were not able to include visual results in the main paper, but did provide visual examples in the supplementary material. We will release all visual results on the website.
>
>
> **Q3: Motivation for not extracting an explicit shape, but depth maps**
>
> We note that many of the baselines were designed more for novel view synthesis and relighting than for surface reconstruction, including all NeRF-based methods.
> We do observe the 3D shapes extracted from these methods using marching cubes being very noisy, whereas the depth maps rendered using the volume rendering equation often look more reasonable.
> Nevertheless, in the revised version, we have included additional geometry evaluation using Chamfer distance in the paper. Please refer to the general response and the revised paper for details.

---

> > ### Author Response · Authors · 2023-08-21
> > **Response for reviewer dsb3's review (cont.)**
> >
> > **Q4: Discussion on societal impact**
> >
> >
> > Thanks for pointing this out. We have included a discussion about societal impact in the supplementary material (marked in blue), which is also attached below.
> >
> > >The purpose of this work is to facilitate the development of general object inverse rendering methods by contributing a real-world evaluation dataset and benchmark to the research community. We believe there is no direct negative societal impact resulting from this benchmark. The technology of inverse rendering has numerous applications that will have positive societal impact, such as improving the realism of computer graphics for virtual and augmented reality, or aiding in scene understanding for robotics applications.
> > However, if leveraged for malicious intentions, it could create a significant negative impact on society. For instance, extremists might exploit an inverse rendering methods to generate fake imagery, commonly known as `deepfakes', which may lead to the spread of misinformation or even intimidate the public. Nevertheless, we believe these possibilities can be eliminated with sufficient supervision from the governments and the research communities. It is important to note that our work focuses solely on common everyday objects, steering clear of any personal information or hazardous items.
> >
> >
> >
> > **Q5: Do we know the error introduced by the position and size of the chrome ball?**
> >
> > We agree that the chrome ball may introduce inevitable systemic errors. For example, the environment map recovered from the chrome ball reflection can be slightly shifted from the true one received by the object, due to the small disposition between the ball and the object. Therefore, we made our best effort to reduce those errors (e.g. move the chrome ball as close to the object as possible). Note that although we have tried many alternatives to capture the environment maps, the chrome ball consistently appears to be the best option for our scenario due to its flexibility and high fidelity. Nonetheless, we will continue to improve the system for better accuracy.

---

### Official Review · Reviewer_gijK · 2023-07-23

**Rating:** 7
**Confidence:** 3
**Correctness:** Yes
**Clarity:** Yes

**Strengths:**

> + It seems to be the first real-world dataset that supports "shape", "relit image", and "lighting" by providing ground-truth scans, multi-view images, and environment lighting, as summarized in Table 1
> + Clear figures: The figures are easy-to-follow and clearly show the data processing pipeline and benchmark comparisons.
> + Comprehensive experiments: Representative inverse rendering methods are included in the experiments.

**Additional Feedback:**

N/A

**Documentation:**

Yes

**Ethics:**

No ethical concern

**Limitations:**

Yes

**Opportunities For Improvement:**

> + More detailed examples or experiments about the comparison with synthetic datasets: the authors illustrate the drawbacks of synthetic datasets in Line 37 ~ 46. However, no experiment is provided to show the proposed datasets can overcome all the aforementioned drawbacks.
> + Why use NVDiffRec for camera pose optimization: NVDiffrec is not designed to optimize the camera pose. Thus it is necessary to illustrate the required modification to NVDiffRec pipeline in this work. However, the authors only mentioned that "for each data point, we try multiple runs with different loss weights and learning rates, and select the set of parameters for the best performance in terms of PSNR on the test images" in supp. More details on how to tune the NVDiffRec pipeline (including hyperparam settings) will benefit the community. Also, since NVDiffRec is also a method to be benchmarked, is it fair to use it for camera pose optimization?
> + Limited objects: as mentioned by the authors, the current version"does not cover translucent objects". Based on the illustrated data collection pipeline, it seems difficult for other researchers to contribute to the dataset without the same pipeline setup. How to scale the dataset to more objects may be an issue.

**Relation To Prior Work:**

Yes

**Summary And Contributions:**

The authors introduce a new real-world 3D Object Inverse Rendering Benchmark. They first identify that existing real-world datasets are not sufficient for evaluating the quality of the material recovery and object relighting. Thus, they introduce a new dataset captured under a variety of natural scenes with ground-truth scans, multi-view images, and environment lighting.

---

> ### Author Response · Authors · 2023-08-21
> **Response for reviewer gijk's review**
>
> **Q1: Comparison with synthetic datasets**
>
> The intention of this real-world benchmark is not to replace synthetic benchmarks, and we believe the latter has great potential as rendering engines continue to improve. Rather, this benchmark is designed to compliment synthetic benchmarks, with a strong focus on sophisticated and challenging real-world scenarios.
>
> We highlight two characteristics of our real-world dataset that are not present in synthetic ones.
> 1. *Inclusion of natural capturing noises.* Our captured scenes consist of various combinations of indoor and outdoor, daytime and nighttime environments and well represent the natural capturing noises that one would typically encounter in real-world applications. It is more difficult for a synthetic rendering engine to reproduce these real-world challenges than simply capturing them.
>
> 2. *Inclusion of real-world reflection effects.* For some of the training data with highly shiny objects, the reflection of the camera and the cart (shown in Figure 4)  are also captured in the image. In some views, the camera also casts shadow on the object and changes the illumination effects. These shading effects can hardly be shown in synthetic datasets, and are definitely worth taking into account in a benchmark as they pose additional challenges for the baselines.
>
> In this work, we have made a significant effort in evaluating a wide range of representative existing methods on our benchmark.
> Despite that, unfortunately, we believe it is extremely difficult to establish a comprehensive side-by-side comparison between our benchmark and the existing synthetic benchmarks, as the objects and scenes are very different, and different methods opt to evaluate on different datasets.
> Nevertheless, we hope having this standard benchmark available will be a good starting point and that valuable insights will continue to emerge as more and more methods begin to evaluate on the same benchmarks.
>
>
> **Q2: Camera pose refinement using NVDiffRec**
>
> We use NVDiffRec to refine the camera poses because it is a highly performant full inverse rendering pipeline that directly supports our scanned meshes as input.
> Specifically, we use all images (training+testing) to train the same NVDiffRec model with the scanned mesh multiple times. Each time the model is assigned with an unique set of camera regularization weights (introduced in NeRF– to regularize the camera deformations). We keep the camera poses optimized from the run that achieves the best PSNR scores on the synthesized views.
> More generally, to best demonstrate the full details of our pipeline, all code for data preparation will be released so that the research community can collectively contribute to the dataset.
>
>
> **Q3: Would the NVDiffRec baseline being benchmarked benefit from the camera pose refinement?**
>
> The camera pose refinement makes use of the high-quality 3D scans and pseudo surface materials recovered from the light box captures and results in very accurate poses.
> All methods being benchmarked including NVDiffRec optimize/predict geometry, material and lighting from the raw input images.
> We do not think this pose refinement step will additionally benefit the benchmarking  performance for NVDiffRec.
>
>
> **Q4: Contribution to the dataset from other researchers**
>
> We thank the reviewer for bringing this up. This is a very valuable suggestion. We plan to create a set of detailed instructions for capturing the data which will allow other researchers to contribute to the dataset.
> The hardware involved in the capturing process is pretty standard and lightweight.
> We will also be actively maintaining and expanding the benchmark, and would be more than happy to work closely with other researchers who are interested in contributing to the dataset.

---

### Official Review · Reviewer_uqj7 · 2023-07-31
**Small, but well-made real-world dataset benchmark for inverse rendering and novel view synthesis**

**Rating:** 7
**Confidence:** 4
**Clarity:** The paper is well written and structu…

**Strengths:**

- The paper makes a good effort of creating a realistic dataset while ensuring the availability of GT data.
I particularly appreciate the effort to go beyond a fully controlled studio setup by additionally capturing objects

- The paper presents an elaborate evaluation of various baseline methods on the proposed dataset.



**Additional Feedback:**

Minor: Why was further Laplacian smoothing (L168) necessary since the 3D models should already be of high quality with little noise given the professional scanning setup?

**Correctness:**

The proposed methodology for creating the dataset and its evaluation is plausible and sound.

**Documentation:**

The overview dataset documentation provided in supplementary material is extensive.
There is not (yet) much detailed documentation about the dataset on the website.

**Ethics:**

I do not see any ethical issues with this work.

**Limitations:**

Limitations are explicitly discussed the paper.
Societal impact is not discussed, but I do not expect any negative impact from this work.


**Opportunities For Improvement:**

- The dataset is with 12 objects rather small and does not comprise a wide variety of object shapes and surface properties.
The dataset size seems minimal, e.g. almost no object has high-frequency geometric details, few objects have more complex topology or geometry, e.g. thin geometry.

While it might be sufficient for evaluation purposes, the dataset seems too small to be used for training purposes.
Since the largest effort seems to set up all the hardware and software pipeline. It seems that using a larger set of objects would have been a good investment with moderate additional effort.


- The capture setup with a fixed lights and a turntable does not allow to fully capture the environment.
Furthermore, due to the single camera + turntable setup the objects are only captured along a ring with view changes only along single axis rotations.
Hence, the novel synthesis evaluation possibilities are limited.


- L116 - Geometry estimation benchmark: The datasets provides high-quality GT geometry for all objects, however, this paragraph states that no geometric evaluation is performed.
Although it is debatable how to optimally assess the geometry of neural representations, there are plenty of established evaluation methods that could be used (e.g. sampling of surface points, iso-surface extraction, depth map and normal rendering extraction).
Since Table 2 shows geometry evaluation with at least depth and normals, this paragraph should be adapted accordingly.


- Sec. 4.2: The benchmark section describes the setup and measures well and all results are well summarized in Tab. 2 and Fig. 6, but there is no discussion of the results at all, which would be a nice addition to the paper, at least some key take-away messages.


**Relation To Prior Work:**

Prior work is well discussed.

**Summary And Contributions:**

The paper presents a dataset of 3D objects for benchmarking inverse rendering and novel view synthesis.
A set of objects is first captured in a controlled studio environment to estimate ground truth geometry, appearance and material properties.
Then the same objects are further scanned in less controlled real-world environments together with the lighting which is captured via a chrome ball.
The paper further presents an elaborate experimental comparison of various baseline methods on the proposed dataset.

---

> ### Author Response · Authors · 2023-08-21
> **Response for reviewer uqj7's review**
>
> **Q1: Object selection and dataset size**
>
> We carefully selected a set of objects covering various combinations of different geometries (ranging from simple shapes _e.g._ cylinders, spheres and cubes to more complex ones _e.g._ teapots and toys) and surface materials (including shiny plastic and ceramics _e.g._ mugs, metal _e.g._ pitcher and coke can, as well as diffuse _e.g._ salt can).
> Note that while some objects appear to have similar shapes, such as `Salt` and `Baking`, they have very different textures, which brings diversity to the benchmarking setup since the inverse rendering methods are designed to solve both geometry and material. In practice, the texture difference indeed leads to drastically different inverse rendering performance.
> For instance, on the brighter `Baking` can, NVDiffRecMC achieves PSNR-HDR scores of 31.0 (view synthesis) / 24.4 (relighting), whereas on the `Salt` can, the scores drop drastically to 17.9 (view synthesis) / 7.5 (relighting), due to the ambiguity of lighting and texture brightness.
> We have included a visual example showcasing this difference in the supplementary material.
> We believe that including these different instances provides a comprehensive benchmarking analysis from various perspectives.
>
> We are continuing to expand object selection and the dataset. Thanks to the suggestions from reviewers `uqj7` and `8Dpe`, we have added to the dataset two new objects captured in both studio setup and 6 “in-the-wild” scenes, 3 for each object: (1) `cactus` (a small potted plant), (2) `curry` (a thin paper box), which further adds to the diversity of the selection. The new collection of objects is visualized in Figure 2 of the revised paper. We have also updated all the evaluation results incorporating these new scenes.
>
> Moreover, we would like to point out that many existing inverse rendering methods are computationally expensive, and that maintaining a reasonable dataset size is crucial to make the benchmark practically useful. For instance, NeRD requires per-scene optimization, and each scene takes 21 hours on one GPU, summing up to a total of 882 GPU hours on the entire benchmark of 42 scenes. We think the current version of the dataset is already suitably sized, ensuring reasonable computation efficiency for benchmark evaluation.
>
> Finally, we agree with suggestions from reviewers `DSB3`, and plan to create a set of detailed instructions for capturing the data which will allow other researchers to contribute to the dataset.
> The hardware involved in the capturing process is pretty standard and lightweight.
> We will also be actively maintaining and expanding the benchmark, and would be more than happy to work closely with other researchers who are interested in contributing to the dataset.
>
>
> **Q2: Fixed lights and a turntable cannot fully capture the environment**
>
> We agree, but note that the only objective of this light box capture is to obtain a set of pseudo surface materials that are used for pose refinement in in-the-wild scenes and a relighting baseline with pseudo GT materials, as explained in `line 169`.
> This simple capture setup can already fulfill the purpose.
> Moreover, the environment in the light box is heavily constrained with nearly uniform white light to make the capture and material decomposition easier.
>
>
> **Q3: Discussion of the results**
>
> Thanks for pointing this out. We have examined the results more carefully and included a discussion of interesting findings in the revised paper (marked in blue). Here, we included a summary of the key takeaways:
>
> - The performance on the geometry and view synthesis benchmark are highly correlated.
> - We observed that IDR obtained better performance on the geometry and view synthesis benchmark compared to NeRF. While NeRF is powerful for fitting complex objects and achieves superior performance in scene reconstruction, it suffers from lower surface reconstruction fidelity compared to SDF-based methods (_e.g._ IDR) which is particularly advantageous in object-centric reconstruction.
> - For the relighting benchmark, NVDiffRecMC achieves the best performance among all methods, including significantly outperforming NVDiffRec, which evidently shows the effectiveness of its differentiable Monte-Carlo-based renderer.
>
> More discussions are included in Section 4.1 of the revised paper.

---

> > ### Author Response · Authors · 2023-08-21
> > **Response for reviewer uqj7's review (cont.)**
> >
> > **Q4: Discussion on societal impact**
> >
> > Thanks for pointing this out. We have included a discussion about societal impact in the supplementary material (marked in blue), which is also attached below.
> >
> > >The purpose of this work is to facilitate the development of general object inverse rendering methods by contributing a real-world evaluation dataset and benchmark to the research community. We believe there is no direct negative societal impact resulting from this benchmark. The technology of inverse rendering has numerous applications that will have positive societal impact, such as improving the realism of computer graphics for virtual and augmented reality, or aiding in scene understanding for robotics applications.
> > However, if leveraged for malicious intentions, it could create a significant negative impact on society. For instance, extremists might exploit an inverse rendering methods to generate fake imagery, commonly known as `deepfakes', which may lead to the spread of misinformation or even intimidate the public. Nevertheless, we believe these possibilities can be eliminated with sufficient supervision from the governments and the research communities. It is important to note that our work focuses solely on common everyday objects, steering clear of any personal information or hazardous items.
> >
> >
> > **Q5: Why Laplacian smoothing on the scans (L168)?**
> >
> > Despite the high quality scanner, we still observed small noise on the surfaces of the scans, and applied minimal Laplacian smoothing following the instructions from the scanner manual.

---

### Author Response · Authors · 2023-08-21
**General response for all reviewers**

We thank the reviewers for the constructive feedback and suggestions. We are glad that reviewers acknowledged the importance of real-world data capture (`uqj7`, `gijK`, `DSB3`, `8Dpe`) and found the benchmark evaluations extensive (`uqj7`, `gijK`).
We have incorporated the reviewers’ suggestions and made a number of changes to the paper (marked in blue):
- We captured 2 additional objects in 6 in-the-wild scenes and included them in the evaluation benchmarks, further expanding the diversity of object selection.
- We further added a direct shape evaluation metric to the geometry benchmark using bi-directional Chamfer distance and reported results in the revised paper.
- We carefully examined the benchmark results and included a detailed discussion in the paper on the strengths and weaknesses of various baseline methods on different tasks and types of scenes, accompanied with more visual examples in the supplementary material.
- We included a discussion on potential societal impact of our work in the revised supplementary material.
- We further expanded the discussion on the limitations of the current benchmark in the supplementary material.

We repeat our response below for a few shared comments from the reviewers, and address specific questions from each reviewer in the individual responses.


**Q1: Object selection and dataset size**

We carefully selected a set of objects covering various combinations of different geometries (ranging from simple shapes _e.g._ cylinders, spheres and cubes to more complex ones _e.g._ teapots and toys) and surface materials (including shiny plastic and ceramics _e.g._ mugs, metal _e.g._ pitcher and coke can, as well as diffuse _e.g._ salt can).
Note that while some objects appear to have similar shapes, such as `Salt` and `Baking`, they have very different textures, which brings diversity to the benchmarking setup since the inverse rendering methods are designed to solve both geometry and material. In practice, the texture difference indeed leads to drastically different inverse rendering performance.
For instance, on the brighter `Baking` can, NVDiffRecMC achieves PSNR-HDR scores of 31.0 (view synthesis) / 24.4 (relighting), whereas on the `Salt` can, the scores drop drastically to 17.9 (view synthesis) / 7.5 (relighting), due to the ambiguity of lighting and texture brightness.
We have included a visual example showcasing this difference in the supplementary material.
We believe that including these different instances provides a comprehensive benchmarking analysis from various perspectives.

We are continuing to expand object selection and the dataset. Thanks to the suggestions from reviewers `uqj7` and `8Dpe`, we have added to the dataset two new objects captured in both studio setup and 6 “in-the-wild” scenes, 3 for each object: (1) `cactus` (a small potted plant), (2) `curry` (a thin paper box), which further adds to the diversity of the selection. The new collection of objects is visualized in Figure 2 of the revised paper. We have also updated all the evaluation results incorporating these new scenes.

Moreover, we would like to point out that many existing inverse rendering methods are computationally expensive, and that maintaining a reasonable dataset size is crucial to make the benchmark practically useful. For instance, NeRD requires per-scene optimization, and each scene takes 21 hours on one GPU, summing up to a total of 882 GPU hours on the entire benchmark of 42 scenes. We think the current version of the dataset is already suitably sized, ensuring reasonable computation efficiency for benchmark evaluation.

Finally, we agree with suggestions from reviewers `DSB3`, and plan to create a set of detailed instructions for capturing the data which will allow other researchers to contribute to the dataset.
The hardware involved in the capturing process is pretty standard and lightweight.
We will also be actively maintaining and expanding the benchmark, and would be more than happy to work closely with other researchers who are interested in contributing to the dataset.


**Q2: Explicit shape evaluation**

Following reviewers’ suggestions (`uqj7`, `DSB3`), we have included numerical evaluation on explicit 3D shapes using the bidirectional Chamfer distance between predicted (extracted) meshes and the ground-truth 3D scans. For methods using an implicit field representation, we apply the standard marching cube algorithm to extract explicit meshes. We follow the implementation from DeepSDF and sample 30000 points from the predicted mesh surface. Results have been included in Table 2 in the revised paper.

We observed that the shape error in general correlates with the depth and normal errors, as expected. IDR obtained lower shape errors compared to NeRF, partially due to larger errors introduced in the marching cube procedure for the latter.

---

> ### Author Response · Authors · 2023-08-21
> **General response for all reviewers (cont.)**
>
> **Q3: Result discussions**
>
> We would like to thank the reviewers for pointing out the lack of discussions on the benchmark results. We have examined the results more carefully and included a discussion of interesting findings in the revised paper (marked in blue). Here, we included a summary of the key takeaways:
>
> - The performance on the geometry and view synthesis benchmark are highly correlated.
> - We observed that IDR obtained better performance on the geometry and view synthesis benchmark compared to NeRF. While NeRF is powerful for fitting complex objects and achieves superior performance in scene reconstruction, it suffers from lower surface reconstruction fidelity compared to SDF-based methods (_e.g._ IDR) which is particularly advantageous in object-centric reconstruction.
> - For the relighting benchmark, NVDiffRecMC achieves the best performance among all methods, including significantly outperforming NVDiffRec, which evidently shows the effectiveness of its differentiable Monte-Carlo-based renderer.
>
> More discussions are included in Section 4.1 of the revised paper.

---

> > ### Comment · Reviewer_8Dpe · 2023-08-21
> > **Specifications of the computation system?**
> >
> > With regard to:
> >
> > "For instance, NeRD requires per-scene optimization, and each scene takes 21 hours on one GPU, summing up to a total of 882 GPU hours on the entire benchmark of 42 scenes."
> >
> > What are the specifications of the computing system being used, e.g., what are the specifications of the GPU, and other components of that impact performance such as e.g. CPU core count and type (if it plays a role in any off-GPU tasks) and bus bandwidth? (I could not find the information in the original/updated versions of the paper/supplementary, but if they exist, I would appreciate being pointed to them). Further in the interest of benchmarking, it would have been beneficial to know what tasks are affecting speed, e.g., is the optimization more compute-intensive, or is it that transferring batches from memory to GPU and results back is a greater contributor toward latency.

---

> > > ### Author Response · Authors · 2023-08-22
> > > **Computation system specifications**
> > >
> > > We trained NeRD with 1 NVIDIA TITAN RTX GPU with 24GB memory and 8 CPU cores (Intel Xeon Gold 5220 CPU @ 2.20GHz). The GPU utilization rate maintained at around 90% during training, suggesting that the compute bottleneck is mostly the model training itself rather than data transfer. Following the default training configurations, it takes 21 hours on each of our scenes with 60 training images of resolution 512 x 512. Note that in the original paper, the training is reported to take 1.5 days on 4 RTX 2080Ti GPUs on one scene (with 62-200 training images depending on the scene).
> > >
> > > Here we briefly recap the optimization procedure of NeRD and explain why the training is slow.
> > > NeRD is a per-scene optimization method for object-centric inverse rendering. Similar to other NeRF-based methods, it represents the scene using two MLP networks (a coarse sampling network followed by a second fine network) that predict the density and appearance parameters at a point in space.
> > > Unlike NeRF, it further decomposes the appearance into a set of BRDF material parameters and environment lighting, and the latter is parametrized using Spherical Gaussians (SG).
> > > During training, for each pixel, it will sample a set of 128 points along the pixel ray and query the MLP networks to obtain the densities and BRDF parameters at each point, which are further integrated along the ray and rendered with the SG lighting to obtain the final pixel color.
> > > This volume rendering process is highly compute-intensive and hence the extended training time.

---

> > > ### Author Response · Authors · 2023-08-28
> > > **Final questions before the discussion ends?**
> > >
> > > We hope our responses have addressed all the reviewers' comments from concerning our submission. As the discussion window closes in a day, if there are any more questions, we would be more than happy to clarify before then.
> > >
> > > Otherwise, we would like thank the reviewers again for their valuable feedback, which has significantly improved the quality and completeness of the benchmark.

---

### Decision · Program_Chairs · 2023-09-22

**Decision:**

Accept (Poster)

**Comment:**

All four reviewers recommend acceptance, with three of them stating it is a good paper (clear accept). The reviewers appreciated the realism of the proposed dataset, the quality of the acquisition, the extensive baseline experiments, and the clarity of the writing. The AC additionally likes the timeliness of the dataset, which seems covering a real need of the community these days.